# A New Remote Hazard and Risk Assessment Framework for Glacial Lakes in the Nepal Himalaya

**David R. Rounce[1], Daene C. McKinney[1], Jonathan M. Lala[1], Alton C. Byers[2], C. Scott Watson[3]**

[1]{Center for Research in Water Resources, University of Texas at Austin, Austin, Texas, USA}

[2]{Institute of Arctic and Alpine Research, University of Colorado Boulder, Boulder, CO, USA}

[3]{School of Geography, University of Leeds, Leeds, LS2 9JT, UK}

Correspondence to: D. R. Rounce (david.rounce@utexas.edu)

Keywords: Glacial lake, GLOF, Debris-Covered Glacier, Hazard, Himalaya, Nepal

## Abstract

Glacial lake outburst floods (GLOFs) pose a significant threat to downstream communities and infrastructure due to their potential to rapidly unleash stored lake water. The most common triggers of these GLOFs are mass movement entering the lake and/or the self-destruction of the terminal moraine due to hydrostatic pressures or a buried ice core. This study initially uses previous qualitative and quantitative assessments to understand the hazards associated with eight glacial lakes in the Nepal Himalaya that are widely considered to be highly dangerous. The previous assessments yield conflicting classifications with respect to each glacial lake, which spurred the development of a new holistic, reproducible, and objective approach based solely on remotely sensed data. This remote hazard assessment analyzes mass movement entering the lake, the stability of the moraine, and lake growth in conjunction with a geometric GLOF to determine the downstream impacts such that the present and future risk associated with each glacial lake may be quantified. The new approach is developed within a hazard, risk, and management action framework with the aim that this remote assessment may guide future field campaigns, modeling efforts, and ultimately risk-mitigation strategies. The remote assessment was found to provide valuable information regarding the hazards faced by each glacial lake and results were discussed within the context of the current state of knowledge to help guide future efforts.

## 1 Introduction

Glacial lake outburst floods (GLOFs) unleash stored lake water often causing enormous devastation downstream that can include high death tolls as well as the destruction of valuable lands and costly infrastructure. As the number and area of glacial lakes continues to increase (Bajracharya and Mool, 2009; Bolch et al., 2011; Gardelle et al., 2011; Carrivick and Tweed, 2013), assessing the risk associated with these potential GLOFs becomes increasingly important. This is especially true in the Himalaya, where a global assessment of the socio-economic impacts resulting from glacier outburst floods found that Nepal and Bhutan had the greatest economic consequences at the national-level despite experiencing fewer floods than other parts of the world (Carrivick and Tweed, 2016). An assessment of previous GLOFs reveals the most common cause of failure in the Himalaya is mass movement (snow, ice, and/or rock) entering the lake (Richardson and Reynolds, 2000; Wang et al., 2012; Emmer and Cochachin, 2013) and subsequently overtopping and eroding the damming moraine. Other triggering mechanisms may include dam settlement and/or piping, the degradation of an ice-cored moraine, the rapid input of water from extreme events, and seismic events (Westoby et al., 2014). Events that occur over a relatively short time period, i.e., minutes to days, such as mass movement entering the lake, intensive rainfall, or intensive snowmelt are referred to as dynamic events, while other events that occur over longer periods of time are referred to as self-destruction or long-term causes (Yamada, 1998; Emmer and Cochachin, 2013). Emmer and Cochachin (2013) highlight the complexity of these self-destructive events as they lump the failure from the degradation of buried ice, hydrostatic pressure, and/or the effects of time all together due to the difficulty of distinguishing the exact cause of failure. Similarly, Richardson and Reynolds (2000) were unable to identify the cause of over 23% of the GLOFs in their study due to a lack of information. This uncertainty regarding other mechanisms of failure stresses the importance of taking a holistic approach towards assessing the hazard of these glacial lakes that accounts for the various triggering mechanisms and the stability of the moraine.

Methods have been developed to characterize the hazard and risk associated with glacial lakes in Cordillera Blanca (Reynolds, 2003; Hegglin and Huggel, 2008; Emmer and Vilímek, 2013; Emmer and Vilímek, 2014), New Zealand (Allen et al., 2009), North America (Clague and

Evans, 2000; O'Connor et al., 2001; McKillop and Clague, 2007a, b), the Swiss Alps (Huggel et al., 2004b; Nussbaumer et al., 2014), the Himalaya (Wang et al., 2008; ICIMOD, 2011; Wang et al., 2012; Fujita et al., 2013; Worni et al., 2013), the Tibetan Plateau (Wang et al., 2011), and other parts of high mountain Asia (Bolch et al., 2011; Mergili and Schneider, 2011). These methods vary considerably based on the parameters considered, the level of importance placed upon each parameter, the amount and type of required input data, their ability to be transferred to other regions, and their levels of objectivity. Emmer and Vilímek (2013) applied a suite of existing hazard assessments to glacial lakes in Cordillera Blanca and found good agreement between the various methods despite the various studies using different parameters, various amounts of qualitative and quantitative information, and being developed for specific regions. This good agreement suggests it may be feasible to accurately classify the hazard of a glacial lake and shows the parameters may be qualitative or quantitative as long as the developed analysis and thresholds are objective.

Unfortunately, the use of various approaches can also lead to different classifications of the hazard associated with an individual glacial lake. Imja Tsho (Tsho is Tibetan for lake), located in the Everest region of Nepal, provides an excellent example of these conflicting classifications. Some studies have stated that Imja Tsho is safe (Watanabe et al., 2009; Fujita et al., 2013), a very low risk (Hambrey et al., 2008), or a moderate risk (Budhathoki et al., 2010). Conversely, ICIMOD (2011) identified Imja Tsho as one of six glacial lakes in Nepal that is a high priority for further investigation and currently remediation efforts to reduce its hazard are under way (UNDP, 2013). These remediation efforts and a similar project completed at Tsho Rolpa (Rana et al., 2000) are excellent steps forward for Nepal with respect to addressing the hazards associated with its glacial lakes. However, it is important that these efforts and resources are properly directed to ensure the development of the most cost-effective, Nepal-specific methods for successfully reducing the risk of glacial lakes as practiced by the Peruvians since the 1950s (Carey, 2010). Conflicting classifications of Imja Tsho suggest relief efforts may be misguided if the lake is indeed safe and also send mixed signals to the general public and downstream communities that these studies are meant to assist. Another good example of these conflicting reports is Chamlang South Tsho (also referred to as West Chamlang), where one study stated the lake was not particularly dangerous (Byers et al., 2013) and another stated the lake was potentially dangerous (Lamsal et al., 2016) despite having access to the same data sets.

These various assessments and conclusions highlight the importance of (1) developing a clear

and straight forward holistic method for assessing the hazard associated with a glacial lake and

(2) constructing a framework that guides management actions for a given glacial lake that

accounts for the various levels of data collection, modeling, and field campaigns. An additional

aspect that is important to incorporate is how the hazard associated with these glacial lakes will

change as the glacier and glacial lakes continue to evolve. The negative mass balance regime

across the central Himalaya (Kääb et al., 2012) will affect the continued growth of existing lakes

and development of new lakes in overdeepened basins (Linsbauer et al., 2016). Anticipating this

future development is especially important when one considers that successful remediation

strategies on hazardous lakes often take more than a decade to secure funding and implement

(Quincey et al., 2007).

This study assesses the performance of existing hazard assessment methods on potentially

dangerous glacial lakes in Nepal to determine how well they agree with one another. These

existing studies are used as the basis for developing a new holistic approach that is objective,

repeatable, and uses readily available information such that it may be applied to any glacial lake.

This approach will be developed within a management action framework such that it may be

used to guide future field campaigns, research, and ultimately risk-mitigation strategies

associated with hazardous glacial lakes.

**2 Study Area**

This study assesses the hazard and risk of GLOFs for eight proglacial lakes in the Nepal

Himalaya (Figure 1). Six of these glacial lakes were identified by ICIMOD (2011) as being high

priority for further investigation: Tsho Rolpa, Lower Barun Tsho, Imja Tsho, Lumding Tsho

(also referred to as Tsho Og), Chamlang South Tsho (also referred to as West Chamlang), and

Thulagi Tsho. The other two glacial lakes considered are Chamlang North Tsho (also referred to

as Lake 464), which was found to be dangerous (Byers et al., 2013) and Dig Tsho, which

experienced a GLOF in 1985 (Vuichard and Zimmermann, 1987). The level of investigation of

each of lake is highly variable.

The three glacial lakes that have received the most attention are Imja Tsho (86°55.5' E, 27°53.9'

30   N, 5007.7 m.a.s.l.), Tsho Rolpa (86°28.6' E, 27°51.7' N, 4527.8 m.a.s.l.), and Thulagi Tsho

(84°29.1' E, 28°29.3' N, 4014.8 m.a.s.l.). ICIMOD (2011) provides a detailed review of the development of these lakes and the field investigations that have been performed to measure the bathymetry, use geophysical techniques to identify the presence of buried ice, and broadly investigate the hazard associated with these lakes. Since this study, the stability of their terminal moraines has been investigated (Fujita et al., 2013) and potential GLOFs have been modeled (Khanal et al., 2015). Additionally, a laboratory experiment was conducted to model the stability of Tsho Rolpa (Shrestha et al., 2013) and other detailed investigations of Imja Tsho's bathymetry (Somos-Valenzuela et al., 2014) and potential floods (Bajracharya et al., 2007; Somos-Valenzuela et al., 2015) have been conducted. Tsho Rolpa is also the only glacial lake in Nepal that has been remediated in an effort to reduce its hazard and subsequently was extensively studied in the field prior to its remediation (Richardson and Reynolds, 2000), while remediation efforts are currently under way for Imja Tsho (UNDP, 2013).

Dig Tsho (86°35.1' E, 27°52.5' N, 4365.9 m.a.s.l.) has been extensively studied and used as a site to assess and improve GLOF models (Cenderelli and Wohl, 2001; Bajracharya et al., 2007; Westoby et al., 2014; Westoby et al., 2015; Watson et al., 2015). Chamlang South Tsho (86°57.5' E, 27°45.3' N, 4951.8 m.a.s.l.) and Chamlang North Tsho (86°57.3' E, 27°47.0' N, 5218.0 m.a.s.l.) were both investigated in the field by Byers et al. (2013), which also included the modeling of a GLOF from Chamlang North Tsho. Chamlang South Tsho has since been further investigated using high resolution digital elevation models (DEMs) and a bathymetric survey (Sawagaki et al, 2012; Lamsal et al., 2016). A field investigation was performed at Lower Barun Tsho (87°05.7' E, 27°47.9' N, 4534.5 m.a.s.l.) in 1993 in connection to potential hydropower systems downstream (Mool et al., 2001), but has received little attention since with the exception of a visual field assessment of the terminal moraine and potential triggers (Byers, 2014). Similarly, Lumding Tsho (86°36.8' E, 27°46.8' N, 4819.3 m.a.s.l.) has received little attention beyond documenting its growth (Bajracharya and Mool, 2009) despite being identified as a high priority glacial lake (ICIMOD, 2011). This study addresses this data gap and shows how a new method and framework may be used to prioritize future glacial lake investigations and management actions.

## 3 Existing Methods

## 3.1 Summary

The first method that was used to assess the hazard of these eight glacial lakes was a "shotgun approach" to determine how their classifications vary using previous qualitative and quantitative assessment methods. The shotgun approach uses the same studies as Emmer and Vilímek (2013) with the exception of Clague and Evans (2000) and Grabs and Hanisch (1993) as these required site specific knowledge that was not possible to obtain from remote sensing. The use of solely remotely sensed data is one of the main goals of the new method and framework developed in this study. The qualitative methods used were O'Connor et al. (2001), Costa and Schuster (1988), and Wang et al. (2008). Wang et al. (2008) highlights ten hazard parameters, but only gives thresholds for eight of them; therefore, the eight parameters with thresholds are used for this qualitative approach. As a specific hazard rating is unable to be determined from these qualitative approaches, the arithmetic mean of the three is used to rank the glacial lakes.

The semi-quantitative method used in the shotgun approach was from Bolch et al. (2011), which was developed for glacial lakes in the Tien Shan using remotely sensed data. Bolch et al. (2011) uses a term called lake area change based on a comparison of lake area to its initial area; however, the initial area of a lake is unclear as they were all small melt ponds at one point in time. Therefore, the lake area change was simplified to give a value of 1 or 0 based on if the lake has grown in the last decade or not, respectively. The quantitative approach of Wang et al. (2011) also used only remotely sensed data with specific thresholds determined from a statistical analysis of 78 lakes in the southeastern Tibetan Plateau. Wang et al. (2011) uses the mean slope of the moraine based on a 100 m buffer around the lake; however, when this was applied to the glacial lakes in this study the mean slope was zero or negative indicating that within the first 100 m of the lake, the moraine is higher than the lake level. Therefore, the steep lakefront area (SLA) from Fujita et al. (2013) was used as a surrogate for the mean slope of moraine so the method could still be applied and account for the slope of the moraine. A more detailed description of these studies may be found in Emmer and Vilímek (2013). Additionally, a new quantitative approach developed by Emmer and Vilímek (2014) for the Cordillera Blanca was used in this study. The approach assesses five GLOF scenarios: (1) moraine overtopping due to mass movement entering the lake, (2) overtopping from a flood upstream, (3) moraine failure

due to mass movement entering the lake, (4) moraine failure from a flood upstream, and (5)

failure resulting from a strong earthquake.  Unfortunately, the locations of seepage points

required for the last scenario were not identifiable from remote sensing, so the dam instability

due to piping was not accounted for.

**3.2 Application of existing methods**

The qualitative hazard assessments show a good deal of variation between the three approaches

(Table 1).  The arithmetic mean reveals the most dangerous lakes based on these three qualitative

approaches are Chamlang North Tsho, Chamlang South Tsho, and Tsho Rolpa, closely followed

by the others with the exception of Imja Tsho and Thulagi Tsho.  Imja Tsho has a lower value

since these methods all use some form of mass movement entering the lake, which currently is

not a threat at Imja Tsho.  Thulagi Tsho is also not susceptible to ice avalanches and by some

methods has a more stable moraine due to the presence of vegetation.  The semi-quantitative and

quantitative assessments give very different classifications of the hazard associated with each

lake (Table 2).  Bolch et al. (2011) emphasizes the size of the lake and its ability to expand, so all

the large glacial lakes that have expanded over the last decade are ranked as high danger (Imja

Tsho, Lower Barun Tsho, Lumding Tsho, Thulagi Tsho, and Tsho Rolpa).  The other glacial

lakes that have already reached their maximum extent are classified as medium danger.  Wang et

al. (2011) is the complete opposite since the glacial lakes that have already reached their

maximum extent are all high risk or very high risk, while the others are medium or low risk.

This conflicting classification is due to the emphasis on parameters associated with mass

movement entering the lake, i.e., both the distance and the slope between the lake and the glacier.

The large glacial lakes that are still expanding have gentle slopes behind their calving fronts,

which cause them to be classified as medium or low.

Emmer and Vilímek (2014) classify all the glacial lakes as highly dangerous as they are all

susceptible to at least one GLOF scenario.  Table 2 shows how many of the five scenarios are a

potential threat to each glacial lake.  The reason for this classification is that the parameters

associated with the mass movement and overtopping scenario are dam freeboard and distance

between the glacier and the lake.  The eight lakes considered in this study all have outlet

channels, so their freeboard by definition is zero.  Furthermore, these eight glacial lakes are

either in contact or within 600 m of their mother glaciers, which results in this methodology

considering them all to be susceptible to this dynamic failure. As Emmer and Vilímek (2014) discussed, the method was developed for scenarios related to Cordillera Blanca such that the framework could be transferred to other regions, but the exact scenarios or parameters used may not be representative of the main threats to other regions. Nonetheless, the assessment yields valuable insight showing Lumding Tsho, Lower Barun Tsho, and Tsho Rolpa have the greatest number of hazard scenarios due to the potential for a flood from a glacial lake upstream, which is important to consider.

The shotgun approach shows that the hazard classification of each glacial lake varies greatly depending upon the selected method, which makes classifying the hazard associated with a particular glacial lake difficult. Fortunately, the shotgun approach is useful as it highlights the most commonly used parameters in previous studies. Table 3 shows the most frequently used parameters are mass movement entering the lake, the moraine width-to-height ratio, the presence of buried ice in the moraine, and the distance between the lake and the glacier. These naturally reflect the most common causes of GLOFs, i.e., mass movement entering the lake and the self-destruction of the moraine due to hydrostatic pressure and/or the degradation of buried ice (Richardson and Reynolds, 2000; Wang et al., 2012; Emmer and Cochachin, 2013). Furthermore, many of the other parameters are simply alternative forms of estimating the potential cause of failure, e.g., glacier snout steepness or the distance between the lake and the glacier are surrogate ways to estimate if the lake is susceptible to an avalanche entering the lake. In this manner, the shotgun approach lends insight into the various parameters or methods that were used to estimate different triggering events. The variety of parameters and the frequency of their use highlight the parameters that are important to consider in addition to highlighting the importance of developing a holistic method that accounts for these various forms of failure.

## 4 New Hazard and Risk Framework

The conflicting hazard classifications from the shotgun approach cast uncertainty on the hazard of each glacial lake that can be confusing and misleading to the stakeholders these studies are meant to assist. Furthermore, they cast uncertainty on which glacial lakes should receive more attention through field campaigns and/or detailed analyses and the specific parameters that should be focused on. This study develops a new hazard and risk assessment framework that is

holistic, objective, reproducible, and initially relies solely on remotely sensed data. Specifically, this framework focuses on two forms of glacial lake failure: dynamic and self-destructive. The term "self-destructive" failure (Yamada, 1998) is used here to avoid any confusion with long term failures resulting from dynamic causes and lake growth, i.e., as a lake grows its expansion may make it susceptible to mass movement entering the lake from areas that could not previously reach the lake. Figure 2 shows the seven parameters used in this study are potential mass movement entering the lake from (1) a snow/ice avalanche, (2) a rockfall, or (3) an upstream flood, (4) the future expansion of the glacial lake, the stability of the moraine based on (5) the hydrostatic pressure and (6) the presence of buried ice, and (7) the downstream impact. These parameters are all estimated using simplistic models and globally available data sets. The approach is referred to as a "remote" hazard assessment. The integration of site-specific field data, high resolution data sets, and more complex models to improve upon this remote assessment will be detailed in the discussion in conjunction with a brief description of the current state of knowledge for each glacial lake investigated.

**4.1 Remote hazard assessment: Methods**

The remote hazard assessment framework is intended to be used as a launching point for assessing the hazards of glacial lakes. The parameters and models used in this framework are all derived from globally available data sets. The DEM utilized in this study is the ASTER GDEM, which is composed of automatically generated DEMs from the Advanced Spaceborne Emission and Reflection radiometer (ASTER) stereo scenes acquired from 2000 to 2008. The ASTER GDEM V2 (hereon referred to as GDEM) has a horizontal resolution of 30 m and a vertical RMSE of ± 15.1 m for mountainous areas (ASTER GDEM Validation Team, 2011). The GDEM was used in this study instead of the SRTM V4 DEM (Farr et al., 2007) due to its higher resolution and its effect on the flood models is discussed in Section 4.1.4. Landsat imagery (Landsat 4/5 TM, Landsat 7 ETM+, and Landsat 8 OLI TIRS) was also used to delineate the glacial lakes and identify areas of snow/ice and land. Landsat imagery was selected for this study as it is the highest resolution multi-spectral imagery that is ubiquitous.

**4.1.1 Mass movement trajectories**

Mass movement entering a glacial lake is a highly hazardous situation for glacial lakes as the entry may cause a tsunami-like displacement wave that can trigger a GLOF. This study

considers three types of dynamic failures: snow/ice avalanches, rockfalls/landslides, and upstream GLOFs (Figure 2). This section describes the mass movement trajectories for avalanches and rockfalls, while the upstream GLOFs are discussed later (Section 4.1.5). Landsat imagery was used to automatically detect glacierized and non-glacierized areas using a ratio of the NIR and SWIR 1 bands with a threshold of 2.2 (Huggel et al., 2004a). In this simplified model, there is no differentiation between snow and ice. Snow/ice avalanche prone areas were identified as any glacierized area with a slope greater than 45° (Alean, 1985), but less than 60° as mass is unlikely to accumulate beyond this threshold (Osti et al., 2011; Shea et al., 2015). Rockfall prone areas were identified as any non-glacierized areas with a slope greater than 30° (Bolch et al., 2011). In this case, the term rockfall is meant to refer to any mass movement from a non-glacierized area and hence is meant to include landslides as well. The lateral moraines were precluded from being rockfall prone areas as they tend to be fairly well developed and are unlikely to lose a large amount of material in a single event. A single flow model using the flow direction algorithm in ArcGIS 10.3 in conjunction with a sink-free GDEM was used to model the path of the mass movement.

The runout distance of the mass movement trajectories was computed using an average slope threshold of 17° and 20° for avalanches and rockfalls, respectively. However, avalanches with a volume less than 6.67 million cubic meters had a higher average slope threshold based on a log relationship between avalanche volume and average trajectory slope (Huggel et al., 2004b)

$$\tan(\alpha) = 1.111 - 0.118 \log(V) \tag{1}$$

The avalanche volume was determined from estimates of avalanche-prone areas and assumed avalanche thickness. Avalanche-prone area was estimated using a variable kernel filter with a 90% threshold to determine the maximum avalanche-prone area with each pixel. The variable kernel filter begins with an individual pixel and determines if 90% of the surrounding pixels including itself, i.e., a 3 x 3 pixel grid, are avalanche-prone. If this condition is satisfied, then the kernel filter increases by one pixel, i.e., a 5 x 5 pixel grid, and continues this process until the 90% threshold fails. The estimated avalanche-prone area is considered to be the largest area that satisfies the 90% threshold. This study assumed three depths for avalanche thickness (10 m, 30

1   m, and 50 m) based on avalanches previously reported in Russia (Huggel et al., 2005) and

common estimates in the Swiss Alps (Huggel et al., 2004) due to a lack of avalanche depth data

in the Himalaya. These assumed thicknesses also agree well with ice thickness estimates derived

from a relationship between thickness, shear stress, slope factor, and slope (Wang et al., 2012).

The ice thickness estimates of the avalanche prone areas based on Wang et al. (2012) ranged

from 15 – 48 m, which lends confidence to the assumed range of values (10 – 50 m) used in this

study. These three scenarios were used in conjunction with the avalanche-prone areas to

estimate avalanche volume. The avalanche volumes ranged from 2.7 x $10^4$ $m^3$ to 6.7 x $10^6$ $m^3$.

**4.1.2 Lake expansion**

Lake growth is crucial to incorporate into hazard assessments as the expansion of a glacial lake

may greatly alter the lake's proximity to potential hazards and increase the volume of water

likely released in a GLOF. Mass movement entering the lake is the most common cause of a

GLOF, so one must determine if dynamic failure is both a current and/or future threat. Multi-

spectral satellite imagery can be used to determine lake expansion rates semi-automatically using

the normalized difference water index (NDWI) (McFeeters, 1996), which is a combination of the

near-infrared (NIR) and blue bands. In the event that the blue band is not available or the

contrast is not clear, the green (Bolch et al., 2008) and/or shortwave infrared (SWIR) bands

(Somos-Valenzuela et al., 2014) may be used as a suitable alternative. Bolch et al. (2008) found

the NDWI method yielded accurate estimates of lake area compared to manual delineations

performed by Bajracharya et al. (2007).

One difficulty associated with the NDWI analysis is the objective selection of the threshold used

to differentiate land and water. Bolch et al. (2011) found the threshold for Landsat images to

range from 0.3 – 0.9 for glacial lakes in northern Tien Shan, but no clear instructions exist for

selecting the threshold for each image and glacial lake. Thakuri et al. (2015) used the same

technique at Imja Tsho and found the lake area to be constant between July and January each

26  year. They suggested this was due to the lake level being constant, but measurements of lake

level were not included. This study uses the same approach with Landsat imagery from 2000 to

2015 captured between September and January each year and assumes the width of the lake

between the lateral moraines is constant based on the findings from Thakuri et al. (2015) and the

assumption of a constant lake level. Two exceptions were made, one for Chamlang North Tsho,

which used an image from May as there were less shadows during this time of year and another for the supplementary image of Lower Barun Tsho in 2008, which used an image from the following April. Additionally, no clear sky, non-banded Landsat imagery was available in 2003, 2004, and 2008 for Lower Barun Tsho, so Advanced Spaceborne Thermal Emission and Reflection Radiometer (ASTER) images between September and January were used for these years instead.

Thresholds were objectively selected such that the average width of the lake was constant between images. This method forces any changes in lake area to be the result of upglacier and/or downglacier expansion, which is the focus of this study. In the event that clouds or the Landsat 7 stripping caused portions of the lake to have gaps in the imagery, a second Landsat image was used to fill in these missing areas using the same criteria. Additionally, large debris-covered icebergs nearing the calving front that could cause pixels to be misclassified as land instead of water were manually corrected in post-processing. The maximum error in the lake area estimates was calculated as the perimeter of the lake multiplied by half the pixel resolution ($\pm$ 15 m for Landsat images).

The expansion rate was estimated as the average rate of areal expansion for all the 10 year intervals available, i.e., if yearly delineations were available from 2000 to 2015, then the six values of the 10 year expansion were averaged to estimate the areal expansion rate. These expansion rates were used in conjunction with ice thickness estimates to determine future lake extents. The ice thickness data used in this study is from the GlabTop2 model (Frey et al., 2014), which estimates the ice thickness from a DEM and glacier outlines. The ice thickness of the glaciers in this study was upwards of 250 m thick behind the calving front of some lakes. The bed topography of the glaciers was computed as the ice thickness subtracted from the surface elevation, which was used to identify potential overdeepenings, i.e., locations in the bed topography that are sinks and allow the lake to expand.

Future lake projections were estimated using the average decadal areal expansion rates over the next 50 years such that the lake's risk to future dynamic failures could be assessed. Glacier flowlines were used to guide the direction of the expansion. The lake level was assumed to be constant over the next 50 years and was estimated as the average elevation of all the lake pixels

based on the lake extent from 2000 as this elevation should be relatively constant over the time

period when DEMs for the GDEM were being captured.

### 4.1.3 Hydrostatic pressure

The exact cause of failure associated with a moraine that spontaneously fails without any

external influence is difficult to pinpoint, but is commonly referred to as "self-destruction"

(Yamada, 1998; Emmer and Cochachin, 2013). One cause of self-destruction is when the

hydrostatic pressure, the pressure a column of water exerts on the moraine, exceeds the structural

capacity of the moraine. Many studies account for the moraine stability using a ratio of the

moraine width-to-height (Table 2), which is subject to large errors if a global DEM is used

(Fujita et al., 2008). Fujita et al. (2013) developed a surrogate parameter known as the steep

lakefront area (SLA) using remotely sensed data that is not as susceptible to the uncertainty

associated with global DEMs. The SLA is the average slope between the lake and any point

within 1000 m of the moraine, which is similar in concept to the mean slope of the moraine

based on a 100 m buffer that was used by Wang et al. (2011). The slope of any point within

1000 m of the moraine is meant to capture the steepest slope between the lake and the base of the

terminal moraine. Fujita et al. (2013) examined moraines that had previously failed and

determined that lakes with a SLA less than 10° were not susceptible to fail.

This study uses the SLA with a threshold of 10° to determine if the hydrostatic pressure may

cause the moraine to self-destruct. NDWI delineations of the lake area are used to identify the

pixels for the SLA calculation. The elevation of these lake pixels was set equal to the average

elevation of the glacial lake from the GDEM based on the year 2000 delineation. Furthermore,

the SLA was only computed for the main body of the lake, i.e., the SLA was not computed from

melt ponds on the damming moraines of the lakes. Initial results of the SLA reported very high

values of the SLA due to the elevation changes between adjacent pixels in the GDEM. These

values are not representative of the slope of the moraine or the hydrostatic pressure on the

moraine, so a 100 m buffer around the lake was used such that the SLA would be accurately

captured.

## 4.1.4 Buried Ice

The other main cause of failure associated with "self-destruction" is the melting of ice within a lake's terminal moraine since a disintegrating ice-core can undermine the structural integrity of the moraine (Richardson and Reynolds, 2000). This can have large implications for the hydrostatic pressure, piping/seepage, and reducing the height of the freeboard associated with the terminal moraine (Emmer and Cochachin, 2013). The importance of accounting for ice-cored moraines is apparent from previous studies (Table 2), but requires detailed information regarding the terminal moraine that is typically not available from remotely sensed data. Bolch et al. (2011) used permafrost as a surrogate parameter to suggest the potential of an ice-cored moraine. A similar approach was assessed in this study using permafrost maps (Gruber, 2012); however, a comparison between lakes with known ice cores and the permafrost maps revealed no correlation. McKillop and Clague (2007) assessed the presence of an ice core according to the shape of the moraine using aerial imagery by assuming that a moraine with a rounded surface with arcuate ridges had an ice core, that a disproportionately large terminal moraine in front of a glacier was potentially ice-cored, and a narrow, sharp-crested moraine with an angular cross-section was ice-free. Unfortunately, this approach is highly subjective and appears to fail for glacial lakes with ice cores in Nepal. For example, Tsho Rolpa has a narrow terminal and lateral moraine that would suggest its moraine is ice free; however, it is known to have an ice core (Yamada, 1998; Richardson and Reynolds, 2000; ICIMOD, 2011).

In the field a common approach to determine if a moraine is ice-cored is by observing ice cliffs or karst topography (Yamada, 1998; Richardson and Reynolds, 2000; ICIMOD, 2011). Another common way is to witness changes in the outlet channel over time, which has been observed at Imja Tsho (Watanabe, 1994), Tsho Rolpa (Yamada, 1998), and Thulagi Tsho (ICIMOD, 2011). This study takes a similar approach using satellite imagery and Google Earth to identify the presence of any water on the moraine or any changes in the outlet. If water is present on the moraine or any changes in the outlet are observed, the moraine is assumed to have an ice core. During the analysis, a combination of Google Earth and satellite imagery was found to help differentiate between shadows and water, but it is recommended to err on the side of caution when one is unsure. Unfortunately, the lack of clear thresholds for identifying water on the moraine or changes in the outlet adds a small amount of subjectiveness to this study. However,

as this was the most effective and least subjective approach for identifying the presence of buried ice in the moraine, it was used for the remote assessment.

### 4.1.5 GLOF Modeling

Flood models play a crucial role in a glacial lake hazard assessment as they identify areas at risk, which allows one to determine the downstream impact. Westoby et al. (2014) provides a thorough overview of the various types of floods and types of models that may be used to reconstruct a GLOF. In short, if the flood entrains enough sediment from the moraine and channel downstream, the flood may transform into a debris flow, which increases the momentum of the flood thereby increasing the GLOF's extent and potential damage. The models used to reconstruct these debris flows or GLOFs range from simple computationally inexpensive GIS-based methods to computationally expensive, physically based numerical models. The GIS-based methods typically rely solely on the geometry of the downstream channel from a DEM. Numerical models have been used to resolve the flow of a GLOF in one or two dimensions. The benefit of two-dimensional models is their ability to capture more complex features and flow characteristics, e.g., multi-directional flows or super-elevation of flow around a channel bed (Westoby et al., 2014). The selection of a particular model or method typically depends on data availability and the desired model complexity.

This study explored the use of two computationally inexpensive flood models: the Modified Single Flow direction (MSF) model developed by Huggel et al. (2003) and the Monte Carlo Least Cost Path (MC-LCP) model developed by Watson et al. (2015). The MSF model is a standard flow direction model that "allows the flow to divert from the steepest descent direction up to 45° on both sides" (Huggel et al., 2003). The model requires a sink-filled DEM, a starting point in the form of a polygon, and a set threshold to stop the model based on the average angle between the starting point and the downstream flood. Depending on the selected threshold, the model can be used to estimate debris flows or flash floods. The MC-LCP model uses a Monte Carlo simulation to vary the DEM as a function of its uncertainty, while identifying the potential flow path for each simulation using a least cost analysis. This model only simulates the flood extent and does not differentiate between debris flows or flash floods.

The lack of data related to previous GLOFs makes it difficult to assess the performance of different models. Watson et al. (2015) used the 1985 GLOF at Dig Tsho to compare the

performance of the MSF and MC-LCP models to the actual flood extents. These models used the GDEM resampled to 15 m and were found to perform reasonably well, although the MC-LCP model had a larger inundation area and fewer artefacts. Watson et al. (2015) also observed that in high relief Himalayan catchments, the requirements for an artificially filled DEM by the MSF model created large linear inundation artefacts, whereas the MC-LCP model displayed improved flow routing and hence is more appropriate for assessing first-order socio-economic impacts of a potential GLOF. It is important to note that the GDEM reflects the post-GLOF terrain, which was severely altered by the GLOF (Vuichard and Zimmermann, 1987). Ideally, a pre-GLOF DEM would be used for model validation so modeled flood extents would not be affected by scouring and deposition in the main channel. Furthermore, the comparison reveals multiple areas where the MC-LCP model does not capture the actual flood extent, which would be highly problematic for a hazard assessment if these areas were populated.

As both the MSF and MC-LCP models have no physical basis, model selection was determined by the one that yielded the most reasonable, conservative estimate of inundation areas when compared to a two-dimensional debris-flow model, FLO-2D, from Imja Tsho (Somos-Valenzuela et al., 2015). Figure 3 shows the flood extent for the FLO-2D, MSF, and MC-LCP models along with the performance of the MC-LCP model for various DEMs (GDEM and SRTM) and resolutions (resampled to 15 or 30 m). The comparison revealed the MSF model (Figure 3B) and the MC-LCP model resampled to a 15 m resolution for the SRTM DEM (Figure 3C) and the GDEM (Figure 3D) severely underestimated the flood extent. On the other hand, the MC-LCP model with the 90 m SRTM DEM yielded too conservative of an estimate (Figure 3G). The 30 m results for both the SRTM DEM (Figure 3E) and GDEM (Figure 3F) agreed well with the FLO2D results (Figure 3A); however, a more detailed analysis revealed the GDEM tracked the main channel better. Therefore, the MC-LCP model with the 30m GDEM was used in this study to model the potential GLOFs from each lake. The GLOF from each lake was routed to the confluence of the Sun Kosi for seven of the eight glacial lakes, and approximately 60 km downstream for Thulagi Tsho based on the assumption that beyond this distance the downstream effects are minimal as the river is able to absorb the flood's energy (Vuichard and Zimmermann, 1987).

## 4.1.6 Downstream Impact

Buildings and agricultural land use data were extracted from the inundation extent of each MC-LCP GLOF scenario to provide a first-pass assessment of socio-economic implications. Buildings were downloaded from OpenStreetMap and validated in Google Earth, which was also used to update the dataset, when necessary, using the most recent imagery. Areas of agricultural land were manually digitized in Google Earth and included all visibly managed land, i.e., land that appeared cleared, walled, farmed, or grazed, and would likely have detrimental socio-economic implications if flooded. The potential downstream impact was broken down into four categories: very high, high, moderate, and low (Table 4). Very high impact is defined as the potential loss of life with no warning (lodges/buildings) and the loss of costly projects or infrastructure (e.g., hydropower). High impact is defined as the potential loss of life with no warning (lodges/buildings) or costly projects (e.g., hydropower). Moderate impact is defined as any damage that is disruptive, which is meant to include damage to agricultural lands, bridges, trails, etc. Lastly, low impact is defined as having no impact on humans, infrastructure, or other projects. For the purpose of this study, buildings are assumed to have permanent occupants whose lives would be threatened in the event of a GLOF. Agricultural lands are considered to be a moderate impact as their occupancy changes temporally depending on the season and in the event that people are in the fields they may be able to hear and/or see an upstream flood and have an opportunity to move to safe ground. The definition of costly projects or infrastructure is fairly subjective, but is meant to refer to any hydropower system or similar project since the loss of a mature hydropower system can effect multiple generations and jeopardize the economic development of the country (Richardson and Reynolds, 2000).

## 4.2 Risk classification and management actions

The hazard elements described above are crucial for determining if a glacial lake is susceptible to a dynamic or a self-destructive failure. Figure 4 shows the workflow that is used to determine if the lake is susceptible to failure and how the cause of failure translates into the hazard associated with the lake. The most dangerous situation is a glacial lake that is susceptible to both dynamic and self-destructive failures, which would classify the lake as a very high hazard. Susceptibility is defined as a hazard greater than low, i.e., a lake that is considered a moderate hazard for dynamic failure and moderate hazard for self-destructive failure is still classified as very high

hazard.  The other scenario that classifies a lake as very high is a lake with a buried ice core that
is susceptible to a snow/ice avalanche as the ice core may alter the height of the moraine over
time and/or the erosion and breach of the moraine.  A lake that is susceptible to avalanches, but
does not have an ice core is classified as high hazard.  Snow/ice avalanches were given the
highest hazard classification since they are the most frequent cause of failure in the Himalaya
(Emmer and Cochachin, 2013).  Additionally, any lake with a buried ice core that is susceptible
to a rockfall, upstream GLOF, or has a steep SLA is classified as high hazard.  These hazard
ratings are meant to reflect self-destructive failures being the second most common cause of
GLOFs followed by mass movement entering the lake from rock or liquid water (Emmer and
Cochachin, 2013).  As temperatures continue to increase, thereby promoting the formation of
more glacial lakes and altering slope stability due to changes in permafrost (Haeberli et al., 2016),
there is a possibility that failures from rockfalls and/or upstream GLOFs may become more
common as they are in Cordillera Blanca (Emmer and Cochachin, 2013).  A lake that does not
have an ice core, but is susceptible to a rockfall, upstream GLOF, or has a steep SLA is
considered to be moderate hazard.

The risk of a GLOF is a function of a glacial lake's hazard and its potential downstream impact
(Figure 5).  This study uses the same ranking scheme as Worni et al. (2013).  Very high risk
lakes are defined as any lake where the downstream impact is very high, i.e., lives and costly
projects are threatened, and the hazard of the lake is very high.  High risk is defined as a lake that
has a high downstream impact and a high hazard, a very high hazard and high/moderate
downstream impact or vice versa.  Moderate risk is defined as a lake that has a moderate hazard
and moderate downstream impact, a moderate/low hazard and high downstream impact or vice
versa, or a low hazard and very high downstream impact or vice versa.  Lastly, low risk refers to
the remaining scenarios that are less of a threat and is defined as lakes that have a low
downstream impact and a low hazard, or a moderate hazard and low downstream impact or vice
versa.  It is important to note that any site that is at risk of a dynamic or self-destructive failure is
valuable from an academic perspective as they may help improve the current state of knowledge
of GLOF hazards.

**5 Results of Remote Hazard Assessment**

The remote hazard assessment builds off of existing knowledge of glacial lakes in Nepal while integrating new approaches to develop a holistic understanding of their hazard and risks. The mass movement trajectories mark the first time these potential triggers have been modeled at a larger scale, which provides valuable information on the potential for mass to enter a lake in addition to identifying the locations of these avalanche and rockfall prone areas that should be further investigated. The stability of a moraine utilizes a previously developed approach, i.e., the SLA (Fujita et al., 2013), in combination with predictions of the presence of buried ice from satellite imagery such that self-destructive failures may be integrated into the hazard framework with the dynamic failures. Similarly, the downstream impacts use a previously developed GLOF model, i.e., MC-LCP approach (Watson et al., 2015), to obtain a conservative estimate of potentially inundated areas downstream. For three of the eight glacial lakes considered in this study, this is the first time their potential GLOFs have been modeled. Furthermore, this study combined lake expansion rates from satellite imagery with ice thickness estimates to model the evolution of these glacial lakes to determine how their hazard may vary over the next 50 years.

Table 5 provides a brief summary of the hazard parameters for each glacial lake. The potential for avalanches and rockfalls should not be surprising as these glacial lakes have developed on avalanche-fed debris-covered glaciers, so their surrounding slopes are commonly unstable. Three of the eight glacial lakes are threatened by a potential upstream GLOF, which makes it important to assess the hazards associated with these upstream lakes as well. The SLA varies between 4.9° at Lower Barun Tsho up to 18.8° for Chamlang North Tsho. Buried ice is predicted to be present in the moraines of five of these lakes, but only two of these glacial lakes have a SLA above the 10° threshold thereby classifying these lakes as a high risk of self-destruction.

The hazard of these eight glacial lakes varies from low to very high with five of the eight lakes currently being very high hazard (Table 6). Four of the lakes classified as very high hazard are susceptible to both dynamic and self-destructive failures. Specifically, Chamlang South Tsho and Tsho Rolpa are both very high hazard for dynamic failure and high hazard for self-destructive failure. The MC-LCP GLOF modeling revealed potential to damage buildings and large swathes of agricultural lands for all eight glacial lakes and potential damage to hydropower projects for two of the lakes. Therefore, the downstream impact for six of the lakes was high and

very high for the other two (Lower Barun Tsho and Thulagi Tsho). The amount of buildings,

agricultural land, and bridges affected varied greatly, which was partly due to differences in the

distance the GLOF was allowed to propagate downstream, but also due to the amount of

development below each glacial lake. Therefore, the inundated buildings per $km^2$ and

percentage of agricultural land affected are beneficial for comparing the impacts of different

glacial lakes. Tsho Rolpa had the highest amount of affected buildings (56.8 buildings $km^{-2}$) and

agricultural land (16.0%) followed closely behind by Thulagi Tsho (49.7 buildings $km^{-2}$ and

13.1%, respectively). Chamlang North Tsho, Chamlang South Tsho, and Lumding Tsho had the

smallest amounts of inundated buildings and agricultural land affected; however, a GLOF from

these lakes could potentially impact on the order of 200 buildings, 2.5 $km^2$ of agricultural land,

and 15 bridges, which is still cause for considerable concern. Based on these hazard

classifications and downstream impacts, the risk for each glacial lake ranged from moderate to

very high. These high risk classifications should not be surprising as all the lakes investigated in

this study (with the exception of Dig Tsho) were previously deemed hazardous or in need of

further investigation (ICIMOD, 2011; Byers et al., 2013).

Figure 6 shows that Lower Barun Tsho, Imja Tsho, and Lumding Tsho continue to expand

rapidly, while Tsho Rolpa and Thulagi Tsho have stagnated in recent years. The other lakes

(Chamlang North Tsho, Chamlang South Tsho, and Dig Tsho) have already detached from their

glaciers and lack the ability to expand. These expansion estimates combined with the mass

movement trajectories reveal that Imja Tsho is susceptible to a dynamic failure in the next 10 –

20 years (Table 5). These results have important implications on the hazard and risk associated

with Imja Tsho as they indicate that in 10 – 20 years Imja Tsho may be a very high hazard and

high risk (Table 6). Additionally, the expansion of Lower Barun Tsho and Lumding Tsho makes

them susceptible to potential avalanches located further upstream; however, this does not alter

their hazard ratings as they are presently at risk as well.

## 6 Discussion

### 6.1 Imja Tsho

Imja Tsho is one of the most well studied glacial lakes in Nepal, yet the remote hazard
assessment still yields new relevant insight. The mass movement trajectories (Figure 7A,B)
show that the lake is currently not at risk of a dynamic failure. These trajectories are
conservative estimates of avalanche and rockfall runout and for a worst-case scenario they
remain at least 800 m and 400 m from the calving front, respectively. The ice thickness behind
the calving front is greater than 200 m thick (Somos-Valenzuela et al., 2014; Frey et al., 2014),
which allows the lake to continue to expand (Figure 7D). A detailed analysis of the growth of
Imja Tsho (Figure 6) shows the lake is expanding at an average rate of $0.032 \pm 0.004$ km$^2$ yr$^{-1}$.
Based on these expansion rates and the ice thickness estimates, the lake may be at risk of a
rockfall entering the lake in 10 years and an avalanche entering the lake in 20 years.

Fortunately, at the present moment the lake is unlikely to have a self-destructive failure as its
SLA is fairly gentle at 6.8°. The melt ponds on the moraine (Figure 7C) suggest the presence of
buried ice, which has been confirmed in the field (Hambrey et al., 2008). The ice-cored moraine
is currently not a large concern as its gentle slope does not produce hydrostatic pressure that
endangers the lake. However, satellite imagery reveals the outlet lakes have been changing
rapidly, which is important to monitor as these changes may alter the hydrostatic pressure on the
moraine (Fujita et al., 2013). If the outlet lakes on the terminal moraine are considered to be part
of the lake, the SLA changes to 11.0° and Imja Tsho would currently be a high hazard and high
risk. Therefore, monitoring the development of the outlet lakes and their influence on the
hydrostatic pressure should be a top priority. The MC-LCP model (Figure 7E,F) shows that 539
buildings and 2.7 km$^2$ of agricultural land are at risk (Table 7), which corresponds to a high
downstream impact.

Imja Tsho is currently classified as a moderate risk due to its low hazard and high downstream
impact. These high downstream impacts highlight the importance of running hydrodynamic
models with high resolution imagery to improve the mapping of inundated areas and inform the
local communities (Somos-Valenzuela et al., 2014). The expansion model reveals that if Imja

Tsho continues to grow at its current pace it will be a very high hazard and high risk in the next 10 – 20 years. Since the expansion is highly concerning, one important area of future work should be measuring the ice thickness upglacier of Imja-Lhotse Shar Glacier using geophysical techniques such that the potential future extent of Imja Tsho may be accurately quantified. Additionally, efforts should focus on modeling the GLOF process chain, i.e., mass movement entering the lake, the wave propagation, the possible breach of the moraine due to the overtopping, and the downstream impacts due to the future risk. The ice-cored moraine has already been well characterized (Hambrey et al., 2008), but is critical to avoid during any lake lowering activities such that a breach is not initiated. Furthermore, while self-destructive failure is not an immediate concern, repeat bathymetric studies on the outlet lakes would provide valuable information regarding the evolution of the moraine to understand how the hydrostatic pressures may change over time. Based on this remote assessment, the current risk mitigation actions to lower the level of the lake (UNDP, 2013), ideally by 20 m (Somos-Valenzuela et al., 2015), are well justified and hopefully will serve as a good example of how to mitigate the risk of a glacial lake before it becomes highly hazardous.

## 6.2 Lumding Tsho

Lumding Tsho was classified as a high priority for further investigation (ICIMOD, 2011), but has received very little attention beyond an analysis of its expansion (Bajracharya and Mool, 2009) and SLA (Fujita et al., 2013). The remote assessment confirms that Lumding Tsho is a very high hazard as it is susceptible to both dynamic and self-destructive failures. The mass movement trajectories reveal the lake is susceptible to avalanches and rockfalls from the southern side slope (Figure S5A,B). Furthermore, the lake is susceptible to a GLOF from Lumding Tsho Teng located 600 m upstream of Lumding Tsho. Lumding Tsho Teng (27°47.4' N, 86°37.3' E, 5141 m a.s.l.) has an area of 0.34 km$^2$ and is susceptible to a large avalanche from its northeastern slope in addition to potential rockfalls from its surrounding slopes. Lumding Teng Tsho also has a SLA of 31.7° indicating its moraine is highly unstable. These measurements on Lumding Teng Tsho indicate the lake is a very high hazard, so an upstream GLOF should be a great concern for Lumding Tsho.

Lumding Tsho also has a SLA (10.5°) that exceeds the stable threshold indicating the lake is susceptible to a self-destructive failure. Fortunately, the lack of ponds and apparently stable

outlet channel on the terminal moraine indicate the moraine is unlikely to be ice-cored (Figure S5C). Detailed analysis of the growth of Lumding Tsho reveals the lake is growing at an average rate of $0.023 \pm 0.002$ km$^2$ yr$^{-1}$ (Figure 6). The lake expansion model estimates the lake may continue to grow 500 m upglacier (Figure S5D), which would make it susceptible to large avalanches (assumed 50 m depth) from Lumding glacier that currently do not reach the lake. Therefore, the lake is likely to become more hazardous in the future as its proximity to more avalanche prone areas increases, but its hazard classification does not change as it is already classified as a very high hazard. The MC-LCP GLOF model (Figure S5E,F) reveals 184 buildings and 2.0 km$^2$ of agricultural land are at risk (Table 7). This downstream impact is the smallest of those considered in this study, but still shows there is considerable risk to livelihoods downstream thereby classifying Lumding Tsho as high risk.

An initial rapid reconnaissance was undertaken from 20 – 24 October 2015 to assess the hazards associated with Lumding Tsho as directed by the management action framework (Figure 5). The short field campaign consisted of a bathymetric study, streamflow measurements of the outlet, and visual inspection of the terminal moraine and surrounding slopes. The bathymetric survey was conducted over two days using an inflatable kayak and a Garmin echoMAP 54dv to measure 4768 points of lake depth. The shoreline was delineated using the NDWI methods as previously reported. The shoreline was converted into point measurements that were used in conjunction with the bathymetric survey to interpolate depth throughout the lake using the Topo to Raster tool in ArcGIS (Somos-Valenzuela et al., 2014). The lake was found to have an average depth of 51 m, a maximum depth of 114 m, and a total volume of $57.7 \times 10^6$ m$^3$ (Figure 8).

Streamflow from the outlet of the lake was measured to be 8.4 m$^3$ s$^{-1}$ in the afternoon of 23 October using a Pygmy Current Meter with an AquaCalc Pro Plus computer (Rickly Hydrological Company). Inspection of the terminal moraine revealed the outlet was the only source of water exiting the lake, i.e., there was no apparent seepage through the terminal moraine. Additionally, no ponds or karst-like features were present on the terminal moraine, which further indicated that the moraine was relatively stable. In contrast to the SLA, the terminal moraine appeared to be relatively gentle and fairly wide (Figure 9C). A detailed analysis of the SLA calculations reveals there is a large waterfall located 1 km downstream of Lumding Tsho. If the distance from the lake used to compute the SLA was changed from 1 km to 950 m, then the SLA changes to 9.8°, which would change the hazard of a potential self-destructive failure from

moderate to low. The waterfall is important to consider as this drop in elevation would generate a significant amount of energy that could exacerbate a GLOF downstream. Therefore, a physically-based flood model that can account for this acquisition of energy would be valuable for improving estimates of the potential downstream impacts.

Visual inspection of the side slopes revealed the slopes were very steep and likely lacked the ability to generate a large rockfall (Figure 9A,B). On the southern slope there was one large boulder that could be a potential concern (highlighted in Figure 9B), but its limited elevation above the lake level would likely cause only a small surge wave. The snow and ice above the southern side slopes was also very steep and no hanging glaciers were apparent. A more detailed assessment of the stability of the side slopes would be invaluable in improving the likelihood of a rockfall or avalanche. Specifically, the three hanging glaciers located behind the calving front (Figure 5.9A) are potential hazards as the lake grows. Physically-based mass movement models would generate important information regarding the size and trajectories of these slopes failures. Unfortunately, due to time constraints, a detailed analysis of Lumding Teng Tsho, the upstream glacial lake, was unable to be conducted. Similar knowledge of the stability of the slopes surrounding Lumding Teng Tsho would inform the likelihood of an upstream GLOF. A bathymetric study on Lumding Teng Tsho and an assessment of the stability of its moraine should also be a high priority. In the event of a GLOF, the northern lateral moraine of Lumding Tsho may or may not protect the lake from the upstream flood. A physically-based GLOF model that accounts for erosion would yield insight into the potential breach of the lateral moraine and subsequent GLOF from Lumding Tsho. Based on the rapid field reconnaissance, the hazard associated with Lumding Tsho can be reduced from very high to high due to the apparent stability of the moraine. However, more detailed analyses of slope stability around Lumding Tsho and Lumding Teng Tsho in addition to modeling the potential GLOFs from both of these lakes should be a top priority.

## 6.3 Chamlang North Tsho

Chamlang North Tsho has already reached its fullest extent, so its proximity to hazards is not going to change over time (Figure S1D). The southern and eastern slopes reveal the lake is very susceptible to rockfalls and avalanches to the extent that any rockfall or avalanche will enter the lake (Figure S1A,B). This potential for a dynamic failure is exacerbated by its steep terminal

moraine, which has a SLA of 18.8°. This SLA is the highest among the eight glacial lakes investigated in this study and much greater than the stability threshold of 10°. Fortunately, no melt ponds were observed on the moraine and the outlet appears to be stable from satellite imagery indicating an ice-cored moraine is unlikely (Figure S1C). Nonetheless, the combination of the steep terminal moraine and the likelihood of mass movement entering the lake give Chamlang North Tsho a classification of very high hazard. The MC-LCP GLOF model (Figure S1E,F) reveals that 244 buildings and 2.5 km$^2$ of agricultural land are at risk (Table 7) causing Chamlang North Tsho to be a high risk.

Based on the remote assessment, the most important area of future investigation should be modeling the GLOF process chain. The avalanche prone slopes were observed by Byers et al. (2013), who identified four overhanging glaciers on Chamlang North Tsho's southern slopes. The use of high resolution imagery may help quantify the size of a potential avalanche. Byers et al. (2013) modeled a potential GLOF using the GDEM with the U.S. Army Corps of Engineers' Hydrologic Engineering Center River Analysis System (HEC-RAS). The ensuing flood model provides an improved estimate compared to the MC-LCP model, which estimates the water level will rise by 9 m at Bung; however, the downstream impacts were not detailed due to the lack of high resolution imagery. The acquisition of high resolution DEMs for this region and/or cross sections at critical locations should be a high priority. This high resolution DEM should be used with two-dimensional flood models to more accurately estimate the downstream impacts. Additionally, a geotechnical survey including sedimentological sampling of the moraine should be performed such that erosion can be properly accounted for in breach scenarios (Westoby et al., 2015) and the stability of the moraine with respect to the current hydrostatic pressures may be quantified. Lastly, a bathymetric survey of Chamlang North Tsho is needed for the high resolution physically-based GLOF modeling.

**6.4 Chamlang South Tsho**

Chamlang South Tsho has very similar hazard characteristics to Chamlang North Tsho. Chamlang South Tsho is no longer expanding as it has already reached its fullest extent (Figure 6). Mass movement trajectories also show that the lake is surrounded by unstable slopes such that any rockfall or avalanche will enter the lake (Figure S2A,B). The SLA exceeds the stability threshold with a value of 10.5° indicating that the moraine is unstable. Additionally, satellite

imagery reveals the presence of melt ponds on the moraine thereby suggesting the moraine is ice-cored (Figure S2C), which has been verified from the differencing of multi-temporal high resolution DEMs (Sawagaki et al., 2012; Lamsal et al., 2016). Therefore, the steep ice-cored moraine is considered to be a high risk to self-destruct. The combination of self-destructive failure and the potential for a dynamic failure classifies this glacial lake as a very high hazard. The MC-LCP GLOF model (Figure S2E,F) also reveals 228 buildings and 2.5 km$^2$ of agricultural land are at risk (Table 7), so Chamlang South Tsho is considered to be a high risk.

The remote assessment of mass movement entering the lake from the surrounding slopes was verified by field observations (Byers et al., 2013) and satellite imagery (Lamsal et al., 2016). Sawagaki et al. (2012) also performed a bathymetric survey on the lake in 2009 and estimated the total volume to be 35.6 x 10$^6$ m$^3$. These detailed field measurements and observations are highly valuable for verifying the remote assessment and furthering the current state of knowledge with regard to the lake; however, these observations have led to two drastically different conclusions from Byers et al. (2013) and Lamsal et al. (2016) regarding the danger of the lake. Byers et al. (2013) concluded the lake was safe from any dynamic failure based on the assumption that any mass movement-generated wave would be dampened and repelled by the length and surficial roughness of the terminal moraine. In direct contrast, Lamsal et al. (2016) stated a large surge wave could easily overtop the ice-cored moraine as it is only 4 – 18 m higher than the lake level, but no calculations were performed to support this assessment. Furthermore, Lamsal et al. (2016) measured the steep slope of the distal face of the terminal moraine, the low elevation of the toe of the terminal moraine, and observed seepage through the terminal moraine, which led them to conclude the lake was dangerous.

These conflicting studies, which had access to the same data and had similar on-site observations, highlight the need to take an objective approach towards understanding the risks associated with a glacial lake. The main priority with respect to Chamlang South Tsho should be modeling the entire GLOF process chain. This objective analysis would clarify the conflicting views on the potential for a dynamic failure. As a first-pass approach, the methods used by Heller and Hager (2009) were applied to estimate the impulse wave height using one of the avalanche volumes and tracks from the mass movement modeling. The modeled avalanche was 8.1 x 10$^4$ m$^3$ (assumed 10 m thickness) located 800 m east of Chamlang South Tsho at an elevation of 5650 m a.s.l. The avalanche density was assumed to be 500 kg m$^3$ and the avalanche was assumed to expand from

its initial width of 90 m to the width of the lake at the location where the avalanche enters the lake (300 m). Based on these assumptions, the avalanche would generate a 20.5 m wave, which would likely easily overtop the ice-cored moraine. This simplified approach highlights the importance of using high resolution satellite imagery to determine the potential sizes of avalanches in conjunction with physically-based models to more accurately model the avalanche, the wave propagation, and any potential overtopping and/or breaching that may occur.

Additionally, the potential for a self-destructive failure needs to be explored in further detail as the remote assessment suggests the lake is unstable, which is supported by the SLA calculations in Fujita et al. (2013) and the observations and measurements from Lamsal et al. (2016). A chemical analysis of the seepage would lend insight as to the source of the water, i.e., whether the water is the melting of the ice core or lake water. Geophysical surveys of the terminal moraine should be used to determine the spatial extent and depth of the ice core and geotechnical surveys of the composition of the debris would be valuable for assessing the stability of the moraine in detail. Lastly, Lamsal et al. (2016) highlighted the potential downstream impact and Byers et al. (2013) noted that a GLOF from Chamlang South Tsho is a concern for these communities, so improved modeling using a physically-based GLOF model should be a top priority. The combination of these modeling efforts and field measurements would definitively determine the hazard of the lake and the threat to downstream communities.

## 6.5 Dig Tsho

Dig Tsho is a prime example of why glacial lakes should not be prioritized based on the size of the lake. In 1985, Dig Tsho was only 0.5 $km^2$ with a maximum depth of 18 m, yet its GLOF had devastating impacts downstream (Vuichard and Zimmermann, 1987). The breach of its moraine reduced the lake to its present size of 0.4 $km^2$ (Figure 6). The remote assessment shows that Dig Tsho is still very susceptible to another dynamic failure (Figure S3A,B) as any mass movement from the surrounding slopes is likely to enter the glacial lake. Fortunately, its previously breached moraine appears to be very stable as its SLA is 8.9° and an ice-core is unlikely (Figure S3C). Based on the potential for another dynamic failure, Dig Tsho is classified as a high hazard. The MC-LCP model (Figure S3E,F) shows that 519 buildings and 2.8 $km^2$ are at risk from a GLOF (Table 7), thereby classifying Dig Tsho as a high risk based on the remote assessment.

Field investigations should assess the current bathymetry of the lake to determine the amount of

water that could be displaced by a GLOF. Most likely the maximum depth greatly diminished

after the GLOF in conjunction with the reduction in the area of the lake, so the potential GLOF

discharge would be smaller than the 1985 GLOF. A 1-D GLOF model that has been applied to

other glacial lakes in Nepal (Byers et al., 2013; Khanal et al., 2015) would be beneficial in

determining how the downstream impacts have changed based on the new bathymetry.

**6.6 Lower Barun Tsho**

Lower Barun Tsho has received little attention despite being considered one of the most

dangerous glacial lakes in Nepal (ICIMOD, 2011). In this regard, the remote assessment yields

valuable information regarding its hazards and can be used to guide future investigations of the

lake. The mass movement trajectories show the lake is very susceptible to rockfalls and

avalanches from its southern slope (Figure S4A,B). Figure 6 also reveals that Lower Barun Tsho

has had the most rapid expansion rate of the eight lakes studied with an average growth of 0.054

$\pm$ 0.006 km$^2$ yr$^{-1}$. The expansion of Lower Barun Tsho places it at further risk of avalanches and

rockfalls located upglacier such that the lake will only be more susceptible to a dynamic failure

in the future (Figure S4D). There also exists a smaller glacial lake located 4.5 km north of

Lower Barun Tsho called Seto Pohkari (27°50' N, 87°5' E, 4842 m a.s.l.). The MC-LCP model

of Seto Pohkari shows that Lower Barun Tsho is at risk of this potential upstream GLOF. Seto

Pohkari has an area of 0.41 km$^2$ and is considered to be a high hazard as the avalanche and

rockfall trajectories reveal the lake is susceptible to mass movement entering the lake from its

surrounding slopes. Fortunately, the moraine of Seto Pohkari appears to be stable with no melt

ponds and a gentle SLA of 4.0°. Similarly, Lower Barun Tsho has a gentle SLA of 4.9°

indicating the lake is not susceptible to a self-destructive failure. However, satellite imagery

reveals there are apparent ponds on the terminal moraine of Lower Barun Tsho (Figure S4C) and

there appear to be changes in its outlet suggesting that Lower Barun Tsho likely has an ice cored

moraine. The combination of the ice cored moraine and the lake's susceptibility to a dynamic

failure classify the lake as a very high hazard. The MC-LCP model (Figure S4E,F) also reveals

that in the event of a GLOF 640 buildings, 5.9 km$^2$ of agricultural land, and potential

hydropower projects (WECS and NEA, 2015) would be at risk (Table 7). Therefore, Lower

Barun Tsho has a very high downstream impact and is classified as a very high risk.

Similar to Lumding Tsho, Lower Barun Tsho should be a main priority of future field campaigns in Nepal as the lake is a very high risk, but has received little attention. Field campaigns should focus on investigating the potential of mass movement entering the lake from the southern slopes of Lower Barun Tsho. This investigation should be coordinated with physically-based modeling efforts of the GLOF process chain to determine how mass movement entering the lake will propagate across the lake and potentially breach the moraine. While the lake is unlikely to fail due to the hydrostatic pressures, a sedimentological survey of the composition of the moraine would greatly improve modeling a potential breach. Geophysical techniques should also be performed on the moraine to determine the presence and spatial extent of the potential ice core as this may have large implications on the breach of the GLOF. The expansion of Lower Barun Tsho is a large concern as it only increases its susceptibility to rockfall and avalanche prone areas upglacier; therefore, geophysical techniques should be used on Barun Glacier to determine the maximum potential extent of the glacial lake. Additionally, bathymetric surveys should be performed on Lower Barun Tsho to aid modeling efforts of the GLOF process chain. Seto Pohkari also requires attention with regard to its bathymetry and modeling the process chain for avalanches and rockfalls from its surrounding slopes. Similar to Lumding Tsho, the northern lateral moraine of Lower Barun Tsho may protect the lake from the upstream GLOF. Therefore, a physically-based flood model for both Lower Barun Tsho and Seto Pohkari would greatly improve the understanding of the risk faced by downstream communities.

**6.7 Thulagi Tsho**

Thulagi Tsho is one of the three glacial lakes where field campaigns were performed to investigate the hazard of a GLOF by ICIMOD (2011). The results from the remote assessment yield valuable information that may be used to supplement these initial field campaigns. Mass movement trajectories reveal the lake is susceptible to rockfalls from both side slopes, but is not susceptible to avalanches (Figure S6A,B). Figure 6 shows that the lake growth has stalled since 2010. The lake expansion model reveals the bed elevation of the glacier behind the calving front is greater than the lake level indicating that the lake may have reached its maximum spatial extent (Figure S6D). An assessment of the terminal moraine shows there are ponds on the terminal moraine and the outlet channel has changed in the last 15 years, which suggests the moraine is ice-cored. Fortunately, the SLA of 7.1° reveals the terminal moraine is apparently

stable. This finding is in direct contrast to Fujita et al. (2013), which estimated a potential flood volume for Thulagi Tsho of 0.6 million $m^3$ indicating the SLA was greater than the 10° threshold. These differences are likely due to the 100 m buffer, the exclusion of the melt ponds, and/or potential differences in the lake delineations and DEMs used in this study. Based on this analysis, the lake is a high hazard due to the chance of a dynamic failure in conjunction with the ice-cored moraine. The MC-LCP GLOF model (Figure S6E,F) reveals 754 buildings, 2.0 $km^2$ of agricultural land, and planned hydropower projects (ICIMOD, 2011) are at risk (Table 7). These downstream impacts were among the most severe of the glacial lakes in this study as shown by the second highest buildings per $km^2$ and percentage of agricultural land. The combination of these very high downstream impacts and high hazard classify Thulagi Tsho as a high risk.

As previously mentioned, the results of the remote assessment provide valuable information to supplement the work performed by ICIMOD (2011) and the one dimensional GLOF modeling performed by Khanal et al. (2015). Specifically, the lake expansion model shows that Thulagi Tsho appears to have reached its maximum extent, which should be confirmed with a geophysical survey measuring ice thickness behind the calving front. Additionally, the mass movement trajectories are the first time any slope stability has been modeled at this site. These trajectories reveal the lake's vulnerability to mass movement entering the lake and should be the focus of future modeling efforts at this lake. High resolution satellite imagery and field inspection should be used to determine the potential size of any rockfall such that these estimates may be applied to a physically-based mass movement model. These mass movement models could be used in conjunction with the bathymetric survey by ICIMOD (2011) to model the wave propagation and breach of the moraine. A sedimentological survey to accompany the geophysical investigations performed by ICIMOD (2011) would allow the moraine stability and breach parameters to be quantified with greater accuracy. Furthermore, Khanal et al. (2015) also found the downstream impacts from a GLOF were very high; therefore, a two-dimensional physically-based model should build off these results to more accurately quantify the risks and vulnerable areas, which may be used to inform the downstream communities.

## 6.8 Tsho Rolpa

Tsho Rolpa is arguably the most well-studied glacial lake in Nepal and currently the only glacial lake that has been remediated (Richardson and Reynolds, 2000). Richardson and Reynolds

(2000) thoroughly discuss the hazards associated with the glacial lake. Nonetheless, the remote assessment yields valuable insight into the future development of the lake and potential vulnerabilities that may guide future work that should be performed on the lake. Similar to Thulagi Tsho, the lake expansion model shows that Tsho Rolpa appears to have reached its maximum extent (Figure S7D), which explains why the lake area has been relatively constant over the last decade. A geophysical survey behind the calving front would be beneficial to support or disprove this finding. If the model is correct, this has large implications on the hazard of the lake as this will limit the magnitude of future avalanches entering the lake, i.e., Richardson and Reynolds (2000) found that the magnitude of avalanches was increasing as the lake grew. The mass movement trajectories show that the lake is susceptible to avalanches from its northern slope (Figure S7A) and rockfalls from its surrounding side slopes (Figure S7B). The avalanche activity has been a major concern for Tsho Rolpa, so the logical next step is to use physically-based models to model the GLOF process chain and determine how vulnerable the lake is to these threats. Satellite imagery from the last decade also reveals changes in the islands near the terminal moraine, which suggest the presence of an ice core (Figure S7C). The ice core has been confirmed and well documented (ICIMOD, 2011). Additionally, the SLA of 17.5° is the second highest of any of the glacial lakes in this study and Fujita et al. (2013) found Tsho Rolpa had the highest potential flood volume of the lakes in their study. The combination of the ice core, the high SLA, and the potential for mass movement to enter the lake confirm the previous assessments that Tsho Rolpa is a very high hazard.

Additionally, there are three glacial lakes located upstream that are threats to Tsho Rolpa. The first upstream glacial lake, Tsho Rolpa Upper 1 (27°50.7' N, 86°27.8' E, 4968 m a.s.l.), is located 1.5 km southeast of Tsho Rolpa and has an area of 0.12 km$^2$. The hazard assessment reveals this upstream glacial lake is susceptible to ice avalanches, rockfalls, and has a SLA of 13.0°, thereby classifying Tsho Rolpa Upper 1 as a high hazard. The MC-LCP model reveals a GLOF has the potential to enter Tsho Rolpa; however, this would require the upstream GLOF to overtop and/or erode the southern lateral moraine of Tsho Rolpa. Similar to Lumding Tsho and Lower Barun Tsho, flood modeling that incorporates erosion would help determine if this lake is a credible threat to Tsho Rolpa. The second glacial lake located upstream of Tsho Rolpa, Tsho Rolpa Upper 2 (27°50.1' N, 86°29.0' E, 4858 m a.s.l.) has an area of 0.03 km$^2$ and is only susceptible to potential rockfalls. It also has a SLA of 10.1° thereby classifying the lake as a

very high hazard.  The third glacial lake, Tsho Rolpa Upper 3 (27°51.4' N, 86°30.0' E, 5316 m

a.s.l.) has an area of 0.02 km$^2$ and is also only susceptible to potential rockfalls.  This lake has a

greater SLA of 20.2°, which also classifies it as a very high hazard.  However, while Tsho Rolpa

Upper 1 is much larger and appears to be a greater hazard than the other two upstream glacial

lakes, the MC-LCP GLOFs reveal a GLOF from Tsho Rolpa Upper 2 or 3 would flow onto

Trakarding Glacier and directly enter Tsho Rolpa.  Therefore, the hazard associated with these

glacial lakes should be further investigated.

The MC-LCP GLOF model of Tsho Rolpa (Figure S7E,F) shows that 2787 buildings, 7.8 km$^2$ of

agricultural land, and 35 bridges would be at risk (Table 7).  These impacts are both the highest

number of buildings per km$^2$ affected and the highest percentage of agricultural land affected,

which classified the risk associated with Tsho Rolpa as very high.  Khanal et al. (2015) used a

one-dimensional model, which confirmed the severe consequences downstream and highlights

the importance of applying a two-dimensional model to create accurate hazard maps for the

communities.  This assessment is in agreement with other studies that Tsho Rolpa is still a high

risk despite the fact that the lake was lowered by 3-4 m.  Those risk-mitigation efforts serve as a

good example that lowering the level of these lakes is possible, but also highlights the need to

lower the lake 15 – 20 m further (Reynolds, 1999).  A detailed analysis of the changes in the

GLOF as a function of the lowered lake level would be a valuable resource to determine the

exact amount the lake should be lowered.

**7 Conclusions**

The remote assessment integrates the key hazard parameters in an objective manner that is

repeatable and relies solely on globally available remotely sensed data.  This study investigated

eight glacial lakes in Nepal that are widely considered to be highly hazardous and was found to

yield valuable insight with respect to each lake regardless of the amount of previous attention the

lake had received.  For Lumding Tsho and Lower Barun Tsho, this was the first time these lakes

have been holistically studied since they were listed as a high priority of further investigation.

For other glacial lakes that have already been studied extensively, e.g., Tsho Rolpa and Imja

Tsho, the remote assessment yielded valuable information regarding their future expansions.

This study is the first of its kind to incorporate detailed modeling of lake growth into a hazard

assessment. For Imja Tsho this is particularly valuable as the assessment is able to identify future hazardous conditions before they occur and hopefully shows the benefit of implementing risk-mitigation strategies prior to the lake becoming highly hazardous.

The remote assessment is meant to be a simple tool for understanding the hazards and is meant to guide the focus of future modeling efforts and field campaigns. The difficulty associated with conducting fieldwork in these areas and the scarcity of site-specific field data required to adequately model the risk at each site, as discussed in this study, highlights the need for coordinated efforts amongst institutions and local agencies to address these knowledge gaps. This collaborative effort is crucial when one considers the variety of expertise that is required to conduct these field campaigns and effectively model the GLOF scenarios. Furthermore, despite the methods in this study only being applied to eight glacial lakes, the framework was developed such that future work may apply the remote assessment to all the glacial lakes in Nepal. In this manner, a holistic and objective understanding of the current and future state of GLOF hazards may be developed.

### *Acknowledgements*

The authors acknowledge the support of the NSF-CNH program (Award #1516912), the USAID Climate Change Resilient Development (CCRD) project, and NASA Goddard Space and Flight Center / UMBC Maryland for the support of Rounce. We also acknowledge the support of Dr. Dhananjay Regmi of Himalayan Research Expeditions for logistical support during fieldwork. The Landsat imagery used in this study was provided by the Land Processes Distributed Active Archive Center (LP DAAC). The base dataset of buildings used in this study is © OpenStreetMap contributors and is licensed under the Creative Commons Attribution-ShareALike 2.0 license.

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

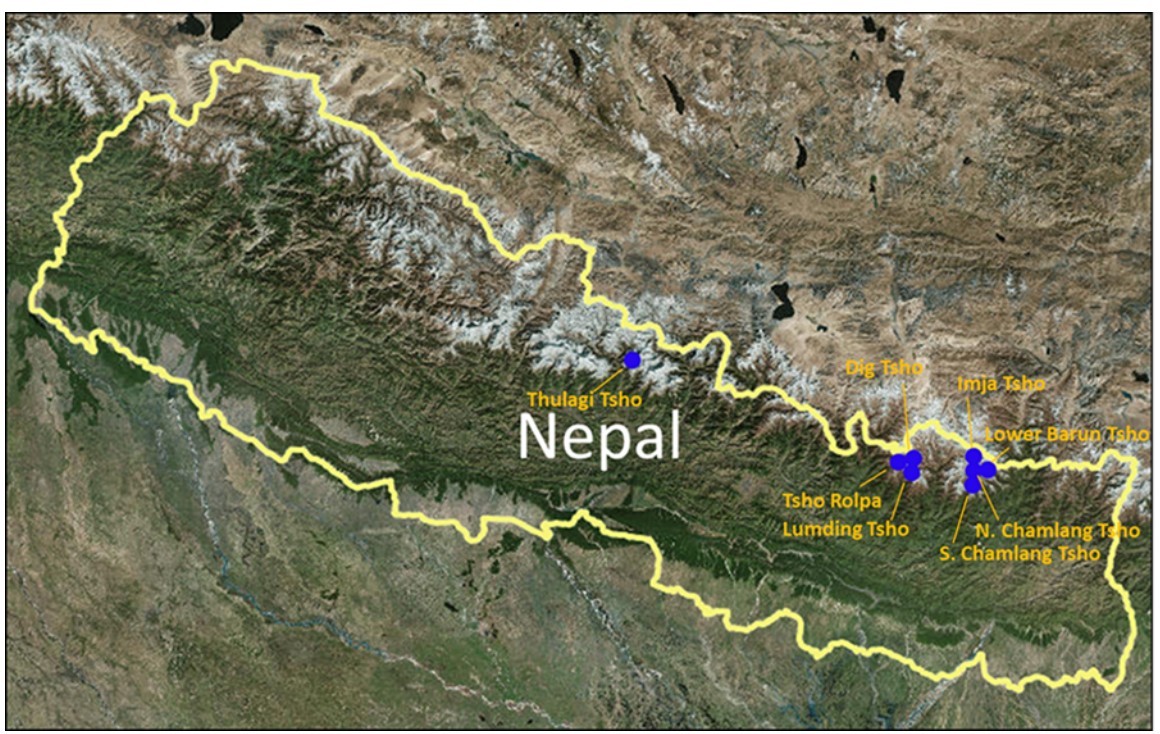

Figure 1. The location of the eight glacial lakes assessed in this study in the Nepal Himalaya

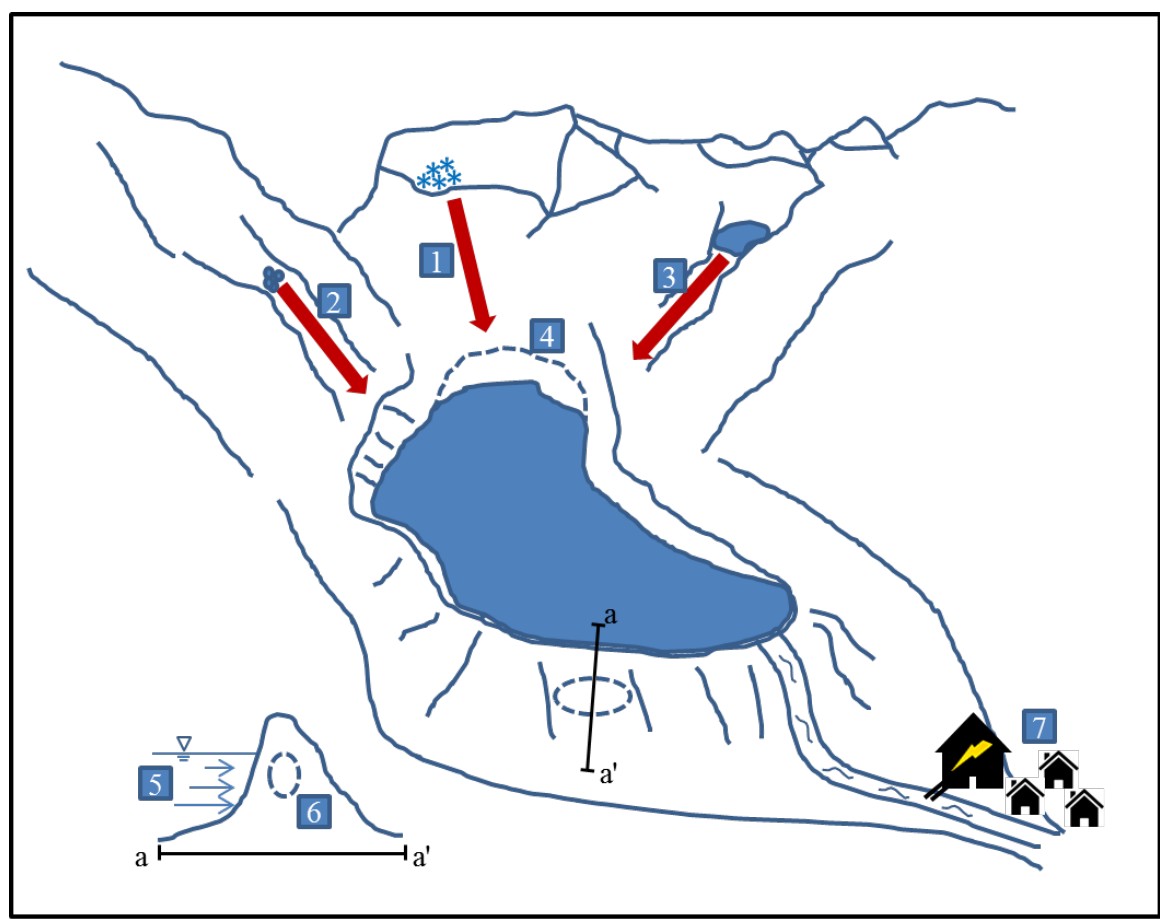

2  Figure 2. Schematic of GLOF hazard parameters used in new method: (1) snow/ice avalanche,

3  (2) rockfall, (3) flood from upstream lake, (4) lake expansion, (5) hydrostatic pressure, (6) ice-

4  cored moraine, and (7) downstream impact.

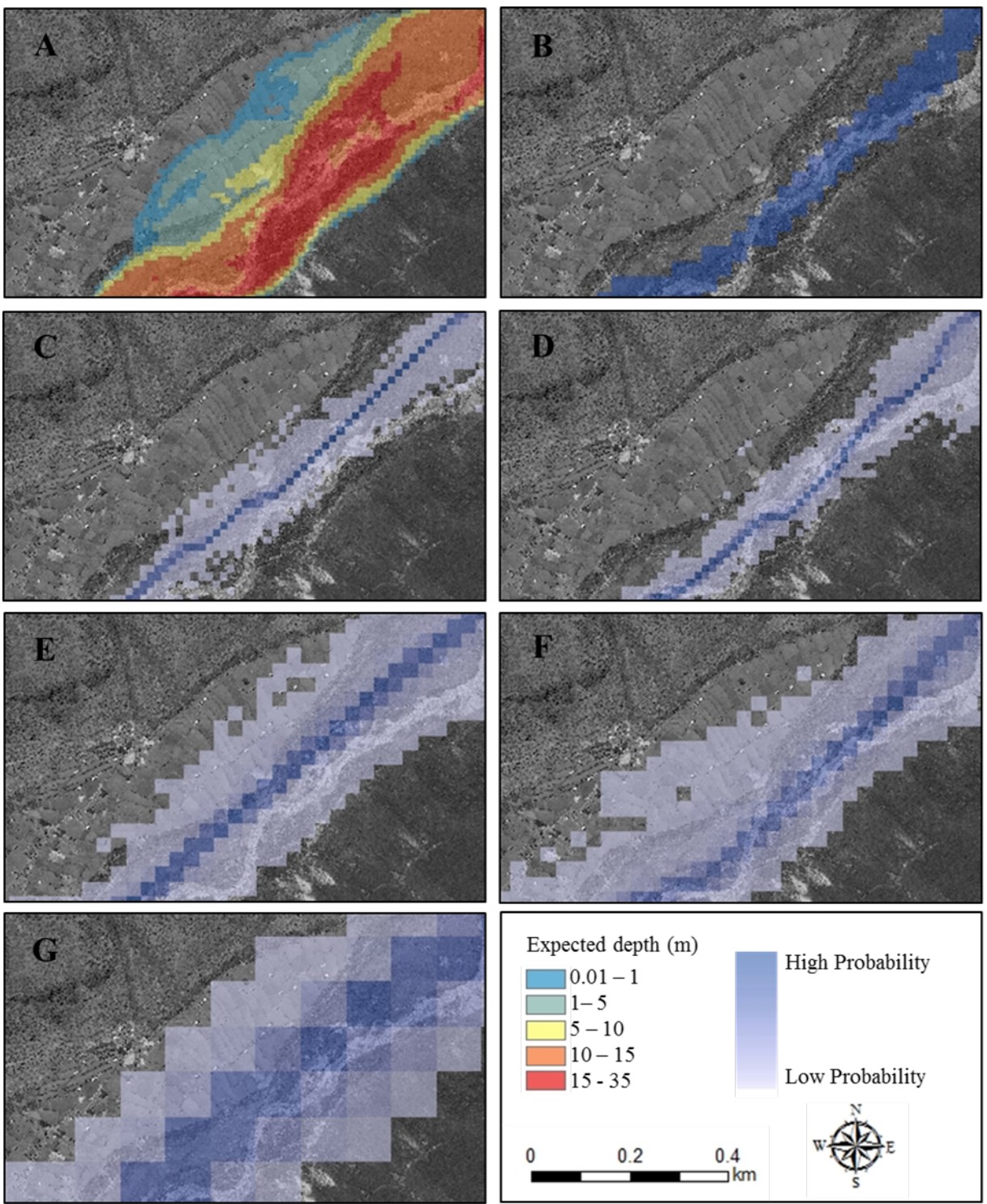

2    Figure 3.  Inundated areas at Dingboche for a GLOF from Imja Tsho using (A) FLO2D (Somos-

3    Valenzuela et al., 2015), (B) the MSF model using the GDEM, (C – G) the MC-LCP model with

4    various DEMs and pixel sizes: (C) SRTM 90 m resampled 15 m, (D) GDEM 30 m resampled 15

5    m, (E) SRTM 90 m resampled 30 m, (F) GDEM 30 m, and (G) SRTM 90 m.

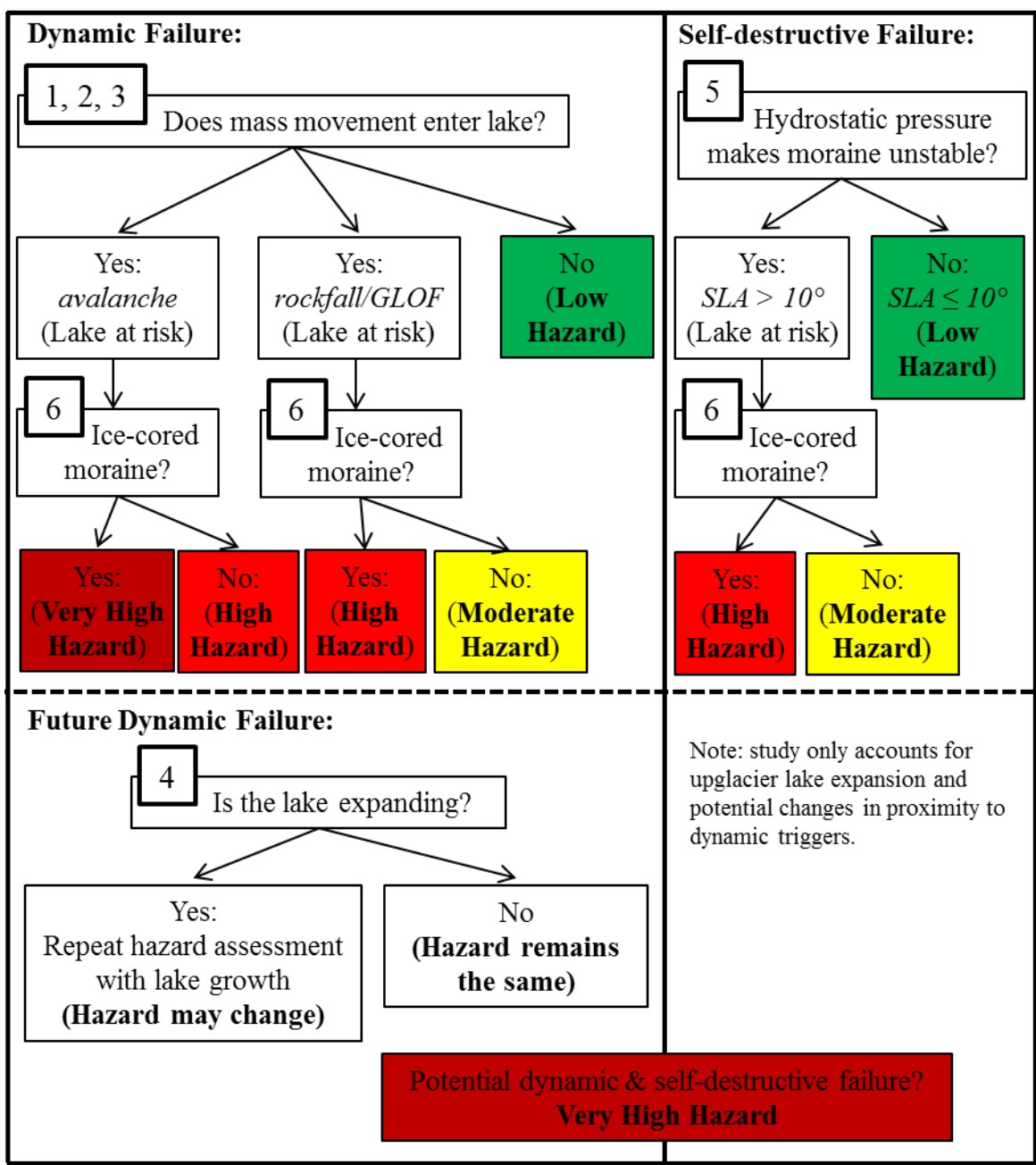

2 Figure 4. Hazard classification flow chart for determining the hazard associated with a glacial
3 lake (numbers refer to hazard parameters in Figure 2).

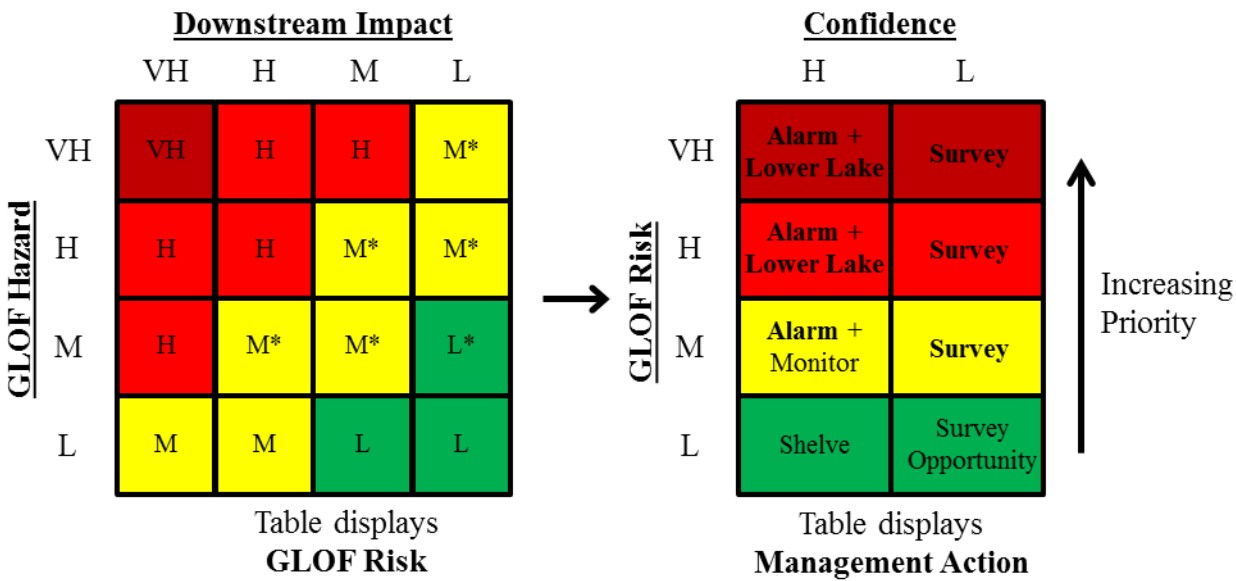

*Site important from academic perspective

2 Figure 5. Risk management and action framework. Left table: GLOF risk is a function of the

3 hazard and downstream impact. Right table: The recommended course of action for a given

4 glacial lake based on the type of assessment.

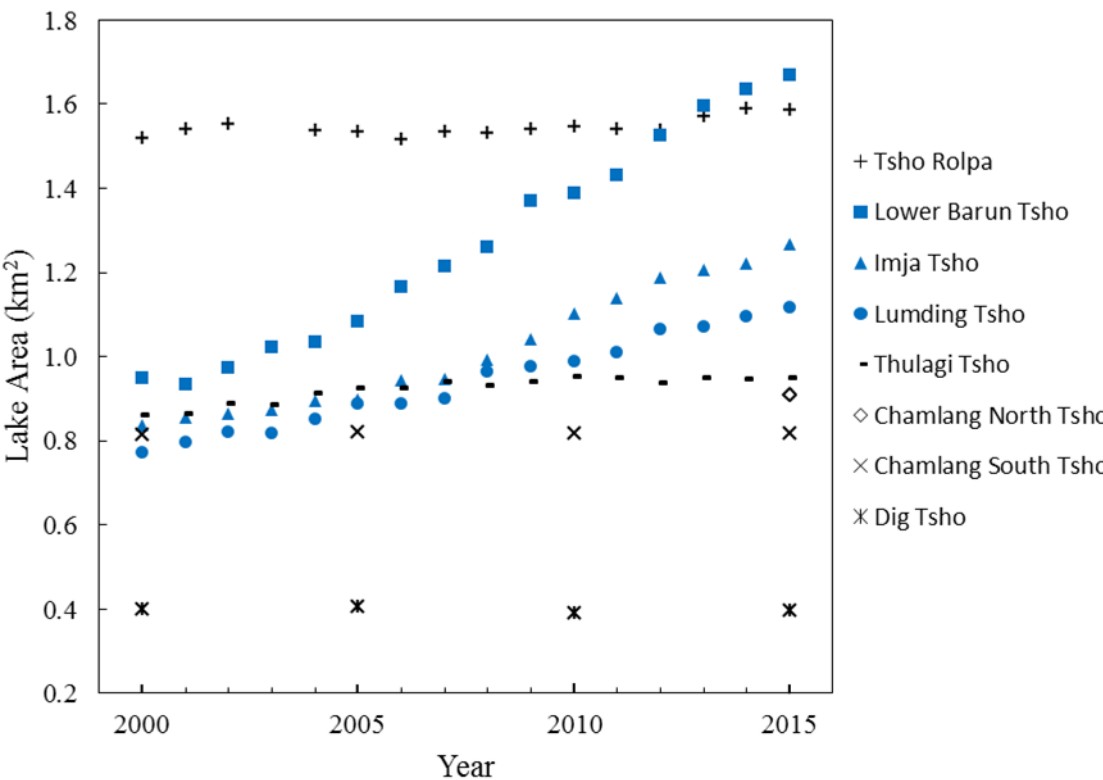

2 Figure 6. Area of glacial lakes derived from satellite imagery using the NDWI method from 2000

3 to 2015 (details in Table S.7).

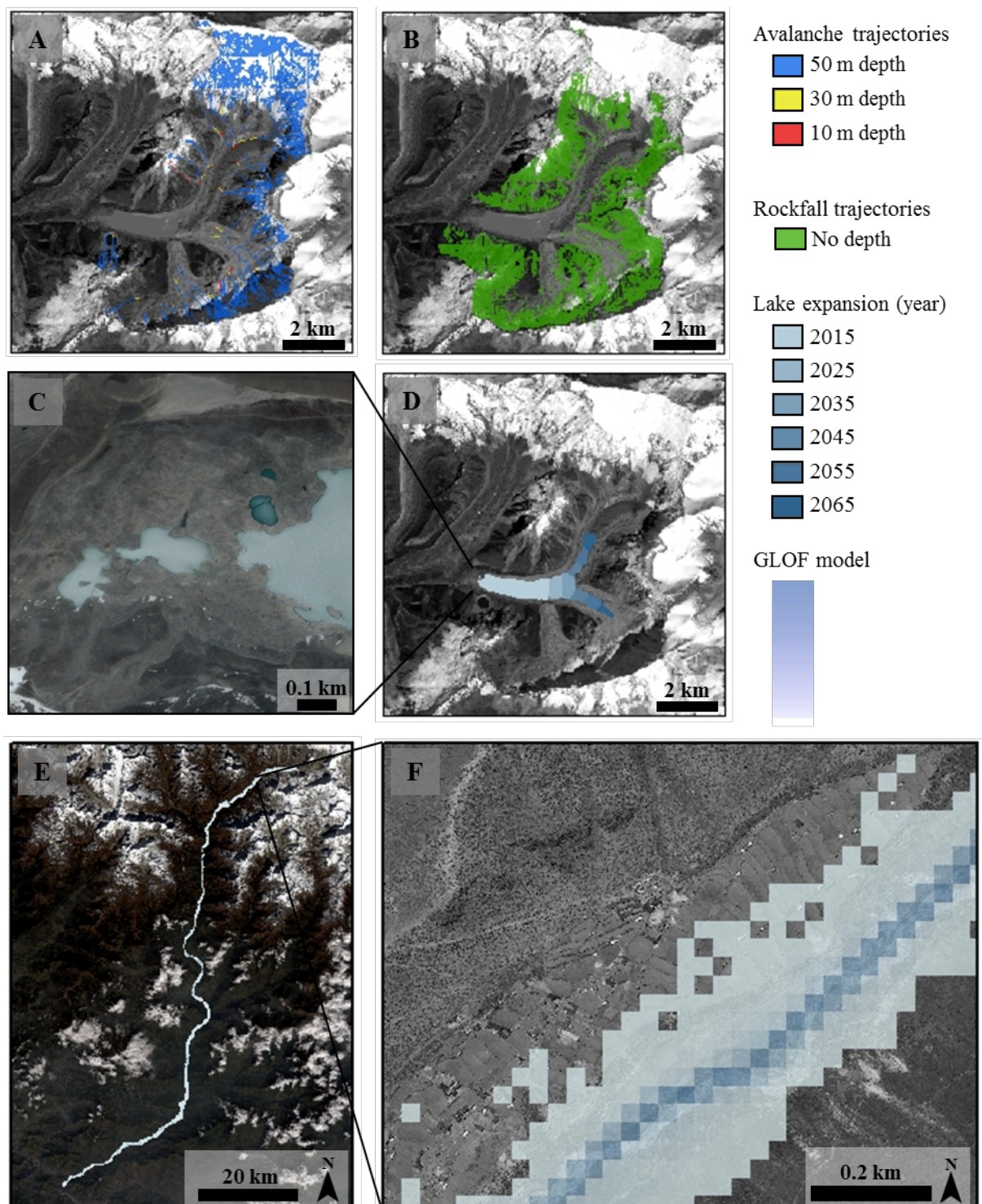

Figure 7. Hazards and downstream impact for Imja Tsho: (A) avalanche trajectories, (B) rockfall trajectories, (C) ponds on the moraine, (D) future lake expansion, and (E) the extent of MC-LCP GLOF model (F) highlighting the impacts at Dingboche.  Background image (A-E) is Landsat 8 from 30 September 2015 and (F) WorldView 2 from 07 June 2015 (DigitalGlobe, Inc.).

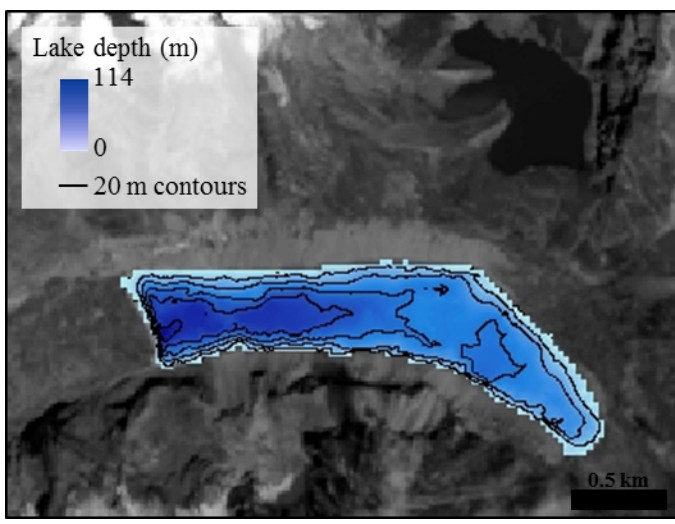

2    Figure 8.  Bathymetric survey conducted on Lumding Tsho on 22-23 October 2015

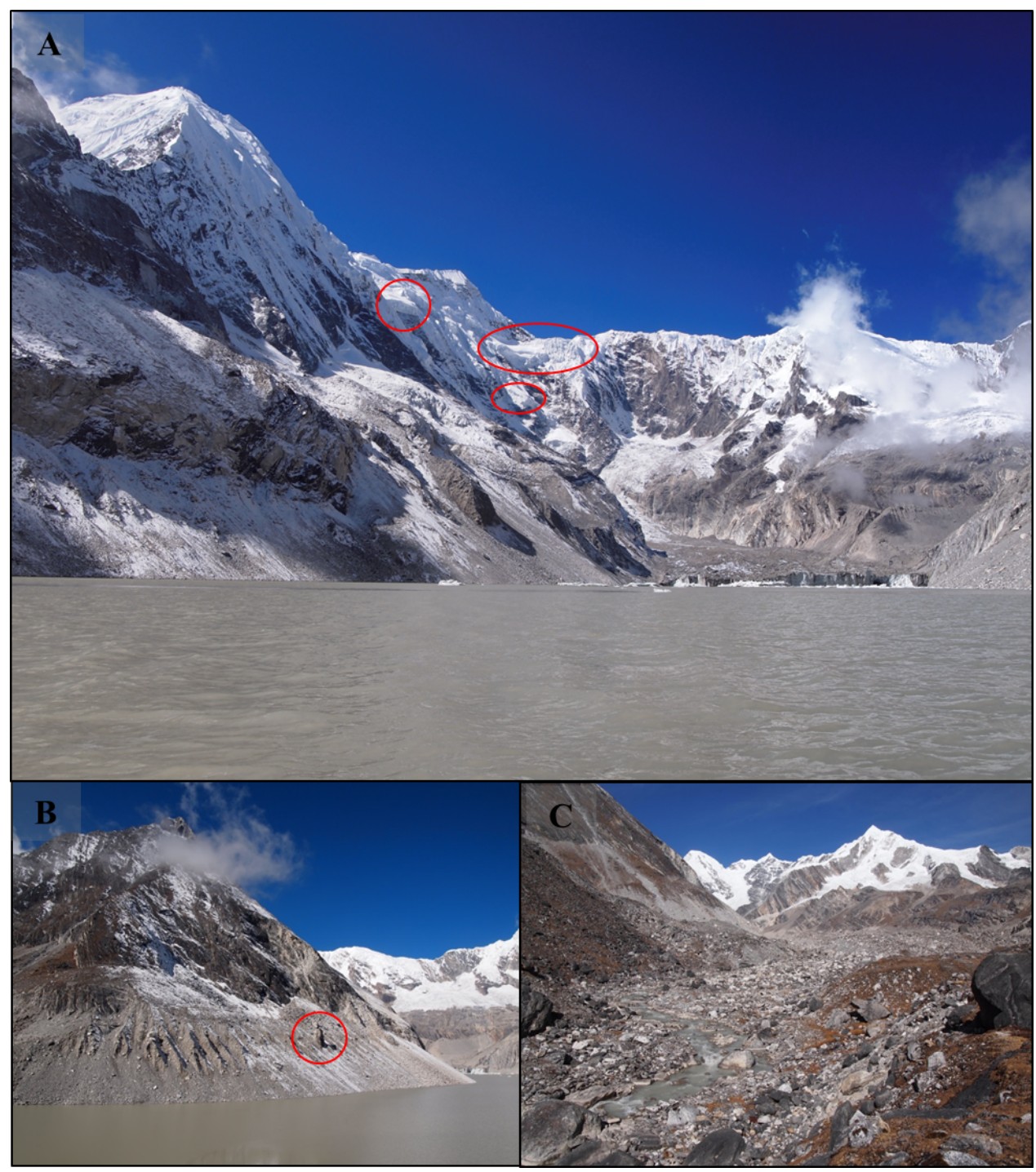

2    Figure 9.  Lumding Tsho and its surrounding slopes with areas of concern highlighted in red

3    showing (A) the calving front and prone areas behind the glacier, (B) the southern side slope, and

4    (C) the terminal moraine and its gentle outlet.

1   Table 1. Previous qualitative hazard assessments applied to eight glacial lakes in Nepal.

2   Fractions are the amount of hazard parameters identified out of the total number of parameters

3   used by each method.  Details shown in Tables S.1 – S.3.

| Lake | O'Connor et al. (2001) | Costa and Schuster (1988) | Wang et al. (2008) | Arithmetic Mean |
|------|-----------------------|---------------------------|--------------------|-----------------|
| Chamlang North Tsho | 2/2 | 3/4 | 6/8 | 0.83 |
| Chamlang South Tsho | 2/2 | 3/4 | 6/8 | 0.83 |
| Dig Tsho | 2/2 | 2/4 | 5/8 | 0.71 |
| Imja Tsho | 1/2 | 2/4 | 5/8 | 0.54 |
| Lower Barun Tsho | 2/2 | 3/4 | 5/8 | 0.79 |
| Lumding Tsho | 2/2 | 2/4 | 6/8 | 0.75 |
| Thulagi Tsho | 1/2 | 2/4 | 5/8 | 0.54 |
| Tsho Rolpa | 2/2 | 3/4 | 6/8 | 0.83 |

6   Table 2. Previous semi-quantitative and quantitative hazard assessments applied to eight glacial

7   lakes in Nepal.  Values and thresholds for classifications are specific to each study.  Fractions are

8   the amount of scenarios for each lake that are considered to be highly dangerous.  Details shown

9   in Tables S.4 – S.6.

| Lake | Semi-Quantitative | | Quantitative | | | |
|------|-------------------|---|--------------|---|---|---|
| | Bolch et al. (2011a) | | Wang et al. (2011) | | Emmer and Vilímek (2014) | |
| Chamlang North Tsho | 0.45 | Medium | 0.85 | Very High | 2/5 | High |
| Chamlang South Tsho | 0.57 | Medium | 0.84 | Very High | 1/5 | High |
| Dig Tsho | 0.41 | Medium | 0.72 | High | 2/5 | High |
| Imja Tsho | 0.63 | High | 0.49 | Low | 1/5 | High |
| Lower Barun Tsho | 0.89 | High | 0.51 | Medium | 3/5 | High |
| Lumding Tsho | 0.79 | High | 0.61 | Medium | 4/5 | High |
| Thulagi Tsho | 0.71 | High | 0.51 | Medium | 1/5 | High |
| Tsho Rolpa | 0.89 | High | 0.60 | Medium | 4/5 | High |

1 Table 3. Most frequently used parameters associated with previous studies (adapted from Emmer

2 and Vilímek, 2013)

| Hazard Parameter | Number of Studies |
|---|---|
| *Stability of Moraine* | |
| Moraine width-to-height ratio | 9 |
| Buried ice in moraine | 8 |
| Piping/seepage through moraine | 7 |
| Dam freeboard | 6 |
| Dam Type | 5 |
| Steepness of moraine | 5 |
| *Potential Triggering Events* | |
| Mass movement into lake | 11 |
| Distance b/w lake and glacier | 8 |
| Glacier snout steepness | 6 |
| Seismis activity | 3 |
| Extreme temp/precip | 3 |
| *Downstream Impact* | |
| GLOF Model | 6 |
| Lake area and/or volume | 5 |

5 Table 4. Descriptions of downstream impact classifications

| Classification | Description of downstream impact |
|---|---|
| Very High | Potential loss of life with no warning (lodges/buildings) *and* the loss of costly infrastructure (e.g., hydropower) |
| High | Potential loss of life with no warning (lodges/buildings) *or* the loss of costly infrastructure (e.g., hydropower) |
| Moderate | Damage that is disruptive, which includes agricultural lands, bridges, trails, etc. |
| Low | No impact on humans, infrastructure, or other projects |

Table 5. Summary of the hazard parameters for the studied glacial lakes.

| Lake | Snow/ice avalanche | Rockfall | GLOF upstream | SLA (°) | Ice-cored moraine | Future change to hazards |
|---|---|---|---|---|---|---|
| Chamlang North Tsho | Yes | Yes | No | 18.8 | No | - |
| Chamlang South Tsho | Yes | Yes | No | 10.5 | Yes[1] | - |
| Dig Tsho | Yes | Yes | No | 8.9 | No | - |
| Imja Tsho | No | No | No | 6.8 | Yes[1,2] | Yes (10-20 yrs) |
| Lower Barun Tsho | Yes | Yes | Yes | 4.9 | Yes[1,2] | Yes (10-20 yrs) |
| Lumding Tsho | Yes | Yes | Yes | 10.3 | No | Yes (10-20 yrs) |
| Thulagi Tsho | No | Yes | No | 7.1 | Yes[1,2] | No change |
| Tsho Rolpa | Yes | Yes | Yes | 17.5 | Yes[2] | No change |

[1]ponds appear to be on the moraine; [2]changes in the outlet area identified

Table 6. Summary of hazard, downstream impact, and risk for each glacial lake.

| Lake | Hazard | | | Downstream Impact | Overall Risk |
|---|---|---|---|---|---|
| | Dynamic | Self-Destructive | Overall | | |
| Chamlang North Tsho | High | Moderate | Very High | High | High |
| Chamlang South Tsho | Very High | High | Very High | High | High |
| Dig Tsho | High | Low | High | High | High |
| Imja Tsho | Low* | Low | Low | High | Moderate* |
| Lower Barun Tsho | Very High | Low | Very High | Very High | Very High |
| Lumding Tsho | High | Moderate | Very High | High | High |
| Thulagi Tsho | High | Low | High | Very High | High |
| Tsho Rolpa | Very High | High | Very High | High | High |

*Future hazard and risk is very high

Table 7. Details of downstream impacts from MC-LCP GLOF models for each glacial lake.

| Lake | Total Area (km$^2$) | Buildings (total) | Buildings (# km$^{-2}$) | Agricultural Land (km$^2$) | Agricultural Land (%) | Bridges (total) | Hydropower Systems |
|---|---|---|---|---|---|---|---|
| Chamlang North Tsho | 28.2 | 244 | 8.7 | 2.5 | 8.8 | 14 | - |
| Chamlang South Tsho | 27.1 | 228 | 8.4 | 2.5 | 9.1 | 14 | - |
| Dig Tsho | 30.2 | 519 | 17.2 | 2.8 | 9.2 | 23 | - |
| Imja Tsho | 32.9 | 539 | 16.4 | 2.7 | 8.1 | 28 | - |
| Lower Barun Tsho | 45.8 | 640 | 14.0 | 5.9 | 12.8 | 20 | Yes |
| Lumding Tsho | 25.5 | 184 | 7.2 | 2.4 | 9.2 | 16 | - |
| Thulagi Tsho | 15.2 | 754 | 49.7 | 2.0 | 13.1 | 20 | Yes |
| Tsho Rolpa | 49.0 | 2787 | 56.8 | 7.8 | 16.0 | 35 | - |

1 **Supplementary Materials**

2 Table S.1.  Results of O'Connor et al. (2001) hazard assessment.

| Lake | Freeboard | Steep glacier calving | Total |
|---|---|---|---|
| Chamlang North Tsho | None | Yes | 2/2 |
| Chamlang South Tsho | None | Yes | 2/2 |
| Dig Tsho | None | Yes | 2/2 |
| Imja Tsho | None | No | 1/2 |
| Lower Barun Tsho | None | Yes | 2/2 |
| Lumding Tsho | None | Yes | 2/2 |
| Thulagi Tsho | None | No | 1/2 |
| Tsho Rolpa | None | Yes | 2/2 |

5 Table S.2.  Results of Costa and Schuster (1988) hazard assessment.

| Lake | Ice-cored moraine | Unstable young moraine, no vegetation | Steep slope moraine (> 40°) | Rock/ice avalanche into lake | Total |
|---|---|---|---|---|---|
| Chamlang North Tsho | No | Yes | Yes | Yes | 3/4 |
| Chamlang South Tsho | Yes | Yes | No | Yes | 3/4 |
| Dig Tsho | No | Yes | No | Yes | 2/4 |
| Imja Tsho | Yes | Yes | No | No | 2/4 |
| Lower Barun Tsho | Yes | Yes | No | Yes | 3/4 |
| Lumding Tsho | No | Yes | No | Yes | 2/4 |
| Thulagi Tsho | Yes | No | No | Yes | 2/4 |
| Tsho Rolpa | Yes | Yes | No | Yes | 3/4 |

8 Table S.3.  Results from Wang et al. (2008) hazard assessment.

| Lake | Top width of dam (< 600 m) | Distal flank steepness* (> 20°) | Ice-cored moraine | Dam width : height (< 2) | Slope of glacier snout (> 8°) | Temp & precip (hot & wet) | Freeboard : dam height (0) | Lake-glacier proximity (< 500 m) | Total |
|---|---|---|---|---|---|---|---|---|---|
| Chamlang North Tsho | 530 | 41.5° | No | 5.4 | 27.4° | Yes | 0 | 0 | 6/8 |
| Chamlang South Tsho | 1050 | 39.7° | Yes | 6.2 | 26.6° | Yes | 0 | 270 | 6/8 |
| Dig Tsho | 460 | 30.1° | No | 7 | 37.6° | Yes | 0 | 960 | 5/8 |
| Imja Tsho | 650 | 31.7° | Yes | 11.1 | 1.7° | Yes | 0 | 0 | 5/8 |
| Lower Barun Tsho | 1000 | 26.7° | Yes | 15 | 7.0° | Yes | 0 | 0 | 5/8 |
| Lumding Tsho | 530 | 31.3° | No | 13.7 | 12.4° | Yes | 0 | 0 | 6/8 |
| Thulagi Tsho | 1000 | 28.1° | Yes | 22.7 | 5.7° | Yes | 0 | 0 | 5/8 |
| Tsho Rolpa | 530 | 39.2° | Yes | 3.4 | 3.0° | Yes | 0 | 0 | 6/8 |

9 *maximum slope on terminal moraine was used

Table S.4. Results of Bolch et al. (2011) hazard assessment

| Lake | Lake area change | Risk of ice avalanche | Rick of rockfall | Ice core | Debris flow | Flash flood | Contact with glacier | Lake area (km$^2$) | Glacier shrinkage | Glacier slope | Stagnant glacier | Hazard | Score |
|---|---|---|---|---|---|---|---|---|---|---|---|---|---|
| Chamlang North Tsho | No | Yes | Yes | No | Yes | - | No | 0.90 | No | 27.4° | No | Medium | 0.45 |
| Chamlang South Tsho | No | Yes | Yes | Yes | Yes | - | No | 0.82 | No | 26.6° | Yes | Medium | 0.57 |
| Dig Tsho | No | Yes | Yes | No | No | Yes | No | 0.40 | No | 37.6° | No | Medium | 0.41 |
| Imja Tsho | Yes | No | No | Yes | No | Yes | Yes | 1.22 | Yes | 1.7° | Yes | High | 0.63 |
| Lower Barun Tsho | Yes | Yes | Yes | Yes | No | Yes | Yes | 1.61 | Yes | 7.0° | Yes | High | 0.89 |
| Lumding Tsho | Yes | Yes | Yes | No | No | Yes | Yes | 1.09 | Yes | 12.4° | Yes | High | 0.79 |
| Thulagi Tsho | Yes | No | Yes | Yes | No | Yes | Yes | 0.95 | Yes | 5.7° | Yes | High | 0.71 |
| Tsho Rolpa | Yes | Yes | Yes | Yes | Yes | - | Yes | 1.59 | No | 3.0° | Yes | High | 0.89 |

Table S.5. Results from Wang et al. (2011) hazard assessment

| Lake | Area of mother glacier (km$^2$) | Distance between lake and glacier (m) | Slope between lake and glacier (°) | Slope of downstream face of dam (°) | Mother glacier snout steepness (°) | Hazard | Score |
|---|---|---|---|---|---|---|---|
| Chamlang South Tsho | 7.5 | 270 | 45 | 10.4 | 26.6 | Very High | 0.84 |
| Chamlang North Tsho | 0.01 | 0 | 27.5 | 12.2 | 27.5 | Very High | 0.85 |
| Dig Tsho | 23.9 | 600 | 31.3 | 8.9 | 37.6 | High | 0.72 |
| Imja Tsho | 2.12 | 0 | 1.7 | 6.8 | 1.7 | Low | 0.49 |
| Lower Barun Lake | 55 | 0 | 7 | 4.9 | 7 | Medium | 0.51 |
| Lumding Lake | 29.1 | 0 | 12.4 | 10.5 | 12.4 | Medium | 0.61 |
| Thulagi Tsho | 56.8 | 0 | 5.7 | 7.1 | 5.7 | Medium | 0.51 |
| Tsho Rolpa | 61.5 | 0 | 3 | 16.4 | 3 | Medium | 0.6 |

Table S.6. Results from Emmer and Vilímek (2014) hazard assessment

| | Chamlang North Tsho | Chamlang South Tsho | Dig Tsho | Imja Tsho | Lower Barun Tsho | Seto Pohkari | Lumding Tsho | Lumding Teng Tsho | Thulagi Tsho | Tsho Rolpa* | Tsho Rolpa Upper1 |
|---|---|---|---|---|---|---|---|---|---|---|---|
| **Terminal Moraine** | | | | | | | | | | | |
| Dam Type | moraine | moraine | moraine | moraine | moraine | moraine | moraine | moraine | moraine | moraine | moraine |
| Dam Freeboard (m) | 0 | 0 | 0 | 0 | 0 | 0 | 0 | 0 | 0 | 0 | 0 |
| Dam Width (m) | 1770 | 1050 | 460 | 750 | 1000 | 900 | 490 | 475 | 1000 | 530 | 500 |
| Dam Height (m) | 330 | 169.8 | 66 | 69 | 66.6 | 50 | 32.3 | 200 | 44 | 165 | 50 |
| Maximum slope of distal face of dam (°) | 40 | 27.4 | 34.6 | 15 | 23.2 | 31.3 | 31.3 | 46.6 | 21.5 | 30 | 39 |
| Remedial work | no | no | no | no | no | no | no | no | no | yes | no |
| **Lake Characteristics** | | | | | | | | | | | |
| Lake Area ($m^2$) | 9.1E+05 | 8.2E+05 | 4.0E+05 | 1.2E+06 | 1.6E+06 | 3.6E+05 | 1.1E+06 | 3.3E+05 | 8.8E+05 | 1.6E+06 | 1.2E+05 |
| Lake Perimeter (m) | 6990 | 4920 | 3480 | 6540 | 8310 | 2850 | 7530 | 3260 | 7230 | 10140 | 2040 |
| Maximum lake width (m) | 650 | 550 | 450 | 690 | 800 | 430 | 510 | 475 | 830 | 600 | 240 |
| Lake Volume ($m^3$) | 3.7E+07 | 3.2E+07 | 1.2E+07 | 7.5E+07 | 8.1E+07 | 9.9E+06 | 4.7E+07 | 9.1E+06 | 3.5E+07 | 8.6E+07 | 2.1E+06 |
| **Lake Surrounding Characteristics** | | | | | | | | | | | |
| Distance b/w lake and glacier (m) | 0 | 200 | 600 | 0 | 0 | 500 | 0 | 500 | 0 | 0 | 240 |
| Width of calving front (m) | 175 | 0 | 0 | | 700 | 0 | 510 | 0 | 575 | 240 | 0 |
| Mean slope b/w lake and glacier (°) | 27.5 | 30.6 | 31.3 | 1.0 | 7.0 | 46.1 | 12.4 | 30.0 | 5.7 | 3.0 | 45.2 |
| Mean slope of last 500m of glacier tongue (°) | 27.5 | 26.6 | 31.3 | 1.0 | 7.0 | 18.8 | 12.4 | 20.0 | 5.7 | 3.0 | 29.1 |
| Max slope of moraine surrounding lake (°) | 74.5 | 73.2 | 67.8 | 54.5 | 70.1 | 57.8 | 66.0 | 56.3 | 68.8 | 57.0 | 67.0 |
| Mean slope of lake surroundings (°) | 34.0 | 45.0 | 45.0 | 37.5 | 45.0 | 35.0 | 35.0 | 30.0 | 45.0 | 45.0 | 35.0 |
| **Results** | | | | | | | | | | | |
| Dam overtopping from mass movement into lake | 1.00 | 0.96 | 0.93 | 1.00 | 1.00 | 0.85 | 1.00 | 0.83 | 1.00 | 1.00 | 1.00 |
| Dam overtopping from upstream GLOF | 0.00 | 0.00 | 0.00 | 0.00 | 0.85 | 0.00 | 0.83 | 0.00 | 0.00 | 1.00 | 0.00 |
| Dam failure from mass movement into lake | 0.64 | 0.46 | 0.57 | 0.21 | 0.39 | 0.52 | 0.52 | 0.73 | 0.37 | 0.50 | 0.63 |
| Dam failure from upstream GLOF | 0.00 | 0.00 | 0.00 | 0.00 | 1.00 | 0.00 | 0.83 | 0.00 | 0.00 | 1.00 | 0.00 |
| Dam failure from strong earthquake | 0.03 | 0.03 | 0.02 | 0.01 | 0.00 | 0.00 | 0.00 | 0.18 | 0.00 | 0.10 | 0.10 |
| **Number of failures lake is susceptible to** | 2/5 | 1/5 | 2/5 | 1/5 | 3/5 | 2/5 | 4/5 | 2/5 | 1/5 | 4/5 | 2/5 |

*Tsho Rolpa has three sizeable glacial lakes upstream, but only one was used for this analysis as it gave the highest results for the upstream GLOF scenarios

1 Table S.7. Details concerning the satellite imagery, dates, bands, thresholds, and area (km$^2$) for
2 each glacial lake in this study.

| Year | Date | Imagery (Bands) | Area (km$^2$) [Threshold] | | | | | | | |
|---|---|---|---|---|---|---|---|---|---|---|
| | | | Chamlang N. Tsho* | Chamlang S. Tsho | Dig Tsho | Imja Tsho | Barun Tsho | Lumding Tsho | Tsho Rolpa | Thulagi Tsho |
| 2000 | 09/12 | L7(1,4) | - | 0.815 [0.68] | 0.400 [0.55] | - | - | 0.773 [0.64] | - | |
| | 09/26 | L7(1,4) | - | - | - | - | - | - | - | 0.860 [0.56] |
| | 09/28 | L7(2,4) | - | - | - | - | 0.949 [0.43] | - | - | |
| | 10/14 | L7(2,4) | - | - | - | 0.835 [0.50] | - | - | - | |
| | 10/14 | L7(1,4) | - | - | - | - | - | - | 1.520 [0.43] | |
| | 10/30 | L7(2,4) | - | - | - | - | Supp [0.25] | - | - | |
| 2001 | 09/15 | L7(2,4) | - | - | - | 0.853 [0.55] | 0.932 [0.45] | - | - | |
| | 09/29 | L7(1,4) | - | - | - | - | - | - | - | 0.864 [0.56] |
| | 10/07 | L7(1,4) | - | - | - | - | - | 0.797 [0.61] | 1.541 [0.43] | |
| | 12/20 | L7(2,4) | - | - | - | - | Supp [0.05] | - | - | |
| 2002 | 10/04 | L7(2,4) | - | - | - | 0.863 [0.35] | 0.974 [0.23] | - | - | |
| | 10/04 | L7(1,4) | - | - | - | - | - | 0.821 [0.41] | 1.553 [0.27] | |
| | 10/20 | L7(2,4) | - | - | - | - | Supp [0.43] | - | - | |
| | 12/05 | L7(1,4) | - | - | - | - | - | - | - | 0.887 [0.35] |
| 2003 | 10/23 | A(1,3) | - | - | - | - | 1.022 [0.00] | - | - | |
| | 11/08 | L7(2,4) | - | - | - | 0.871 [0.50] | - | - | - | |
| | 11/08 | L7(1,4) | - | - | - | - | - | 0.818 [0.64] | - | |
| | 11/14 | L5(1,4) | - | - | - | - | - | - | - | 0.885 [0.54] |
| | 11/16 | L7(1,4) | - | - | - | - | - | Supp [0.60] | - | |
| 2004 | 10/09 | A(1,3) | - | - | - | - | 1.035 [0.00] | - | - | |
| | 10/17 | L5(1,4) | - | - | - | - | - | 0.851 [0.62] | 1.536 [0.44] | |
| | 11/10 | L7(2,4) | - | - | - | 0.893 [0.35] | - | - | - | |
| | 11/16 | L5(1,4) | - | - | - | - | - | - | - | 0.912 [0.51] |
| 2005 | 10/10 | L7(2,4) | - | - | 0.406 [0.12] | - | 1.082 [0.45] | - | - | |
| | 10/28 | L7(2,4) | - | - | - | 0.898 [0.10] | Supp [0.15] | 0.887 [0.20] | - | |
| | 11/03 | L5(1,4) | - | - | - | - | - | - | - | 0.923 [0.48] |
| | 11/05 | L5(1,4) | - | 0.822 [0.63] | - | - | - | Supp [0.34] | 1.535 [0.41] | |
| 2006 | 09/05 | L5(1,4) | - | - | - | - | - | 0.888 [0.57] | 1.517 [0.44] | |
| | 10/15 | A(1,3) | - | - | - | - | 1.165 [0.45] | - | - | |
| | 11/16 | L7(4,5) | - | - | - | 0.943 [0.35] | - | - | - | |
| | 11/22 | L5(1,4) | - | - | - | - | - | - | - | 0.925 [0.50] |
| 2007 | 09/22 | L5(1,4) | - | - | - | - | - | - | - | 0.940 [0.48] |
| | 10/02 | L7(2,4) | - | - | - | 0.947 [0.30] | - | - | - | |
| | 10/02 | L7(1,4) | - | - | - | - | - | Supp [0.41] | Supp [0.25] | |
| | 10/18 | L7(2,4) | - | - | - | - | 1.213 [0.45] | - | - | |
| | 11/03 | L7(2,4) | - | - | - | - | Supp [0.30] | - | - | |
| | 11/19 | L7(1,4) | - | - | - | - | - | 0.900 [0.48] | 1.534 [0.25] | |
| 2008 | 10/12 | L5(1,4) | - | - | - | - | - | 0.965 [0.65] | 1.532 [0.41] | |
| | 10/20 | L7(2,4) | - | - | - | 0.992 [0.30] | - | - | - | |
| | 10/26 | L5(1,4) | - | - | - | - | - | - | - | 0.932 [0.47] |
| | 11/05 | L7(2,4) | - | - | - | - | 1.261 [0.45] | - | - | |
| 2009 | 04/30 | L7(2,4) | - | - | - | - | Supp [0.45] | - | - | |
| | 09/27 | L5(1,4) | - | - | - | - | - | - | - | 0.941 [0.45] |
| | 10/15 | L5(2,4) | - | - | - | - | 1.370 [0.15] | - | - | |
| | 10/15 | L5(1,4) | - | - | - | - | - | 0.977 [0.64] | 1.541 [0.41] | |
| 3 | 10/23 | L7(2,4) | - | - | - | 1.041 [0.30] | - | - | - | |

# Table S.7 (Continued)

| Year | Date | Sensor | | | | | | | | |
|---|---|---|---|---|---|---|---|---|---|---|
| 2010 | 10/10 | L7(1,4) | - | - | 0.390 [0.54] | - | - | Supp [0.65] | Supp [0.36] | |
| | 10/26 | L7(2,4) | - | - | - | 1.101 [0.30] | 1.389 [0.30] | - | - | |
| | 10/26 | L7(1,4) | - | 0.817 [0.48] | - | - | - | 0.989 [0.54] | 1.545 [0.29] | |
| | 12/03 | L5(1,4) | - | - | - | - | - | - | - | 0.952 [0.43] |
| 2011 | 01/14 | L7(2,4) | - | - | - | - | Supp [0.05] | - | - | |
| | 09/03 | L5(1,4) | - | - | - | - | - | - | 1.542 [0.43] | |
| | 10/13 | L7(2,4) | - | - | - | - | 1.432 [0.35] | - | - | |
| | 10/13 | L7(1,4) | - | - | - | - | - | Supp [0.44] | - | |
| | 10/19 | L5(1,4) | - | - | - | - | - | - | - | 0.950 [0.43] |
| | 10/29 | L7(2,4) | - | - | - | 1.139 [0.30] | Supp [0.35] | - | - | |
| | 10/29 | L7(1,4) | - | - | - | - | - | 1.001 [0.45] | - | |
| 2012 | 09/27 | L7(1,4) | - | - | - | - | - | - | - | Supp [0.28] |
| | 09/29 | L7(4,5) | - | - | - | 1.187 [0.10] | - | - | - | |
| | 09/29 | L7(2,4) | - | - | - | - | 1.525 [0.40] | - | Supp [0.18] | |
| | 10/13 | L7(1,4) | - | - | - | - | - | - | - | 0.936 [0.28] |
| | 10/31 | L7(2,4) | - | - | - | - | Supp [0.40] | - | - | |
| | 10/31 | L7(1,4) | - | - | - | - | - | 1.064 [0.44] | - | |
| | 11/16 | L7(1,4) | - | - | - | - | - | Supp [0.21] | 1.538 [0.05] | |
| 2013 | 10/08 | L8(2,5) | - | - | - | - | - | - | - | 0.950 [0.05] |
| | 10/10 | L8(3,5) | - | - | - | 1.206 [0.10] | 1.595 [0.10] | - | - | |
| | 10/10 | L8(2,5) | - | - | - | - | - | 1.071 [0.22] | 1.571 [0.08] | |
| 2014 | 09/27 | L8(3,5) | - | - | - | 1.220 [0.15] | 1.635 [0.08] | - | - | |
| | 09/27 | L8(2,5) | - | - | - | - | - | Supp [0.22] | - | |
| | 10/11 | L8(2,5) | - | - | - | - | - | - | - | 0.947 [0.04] |
| | 11/14 | L8(2,5) | - | - | - | - | - | 1.094 [0.18] | 1.589 [0.05] | |
| 2015 | 05/25 | L8(2,5) | 0.910 [0.05] | - | - | - | - | - | - | |
| | 09/30 | L8(2,5) | - | 0.818 [0.21] | 0.396 [0.14] | 1.265 [0.15] | 1.670 [0.08] | 1.115 [0.20] | 1.588 [0.06] | |
| | 11/15 | L8(2,5) | - | - | - | - | - | - | - | 0.948 [0.04] |

*Required heavy manual editing to account for shadows

Note: Error in lake area as a percentage of lake area ranged from 4 to 14% with an average of 10%.

"Supp" stands for supplementary image used to fill in for clouds or stripping

"L5" stands for Landsat 5

"L7" stands for Landsat 7

"L8" stands for Landsat 8

"A" stands for ASTER

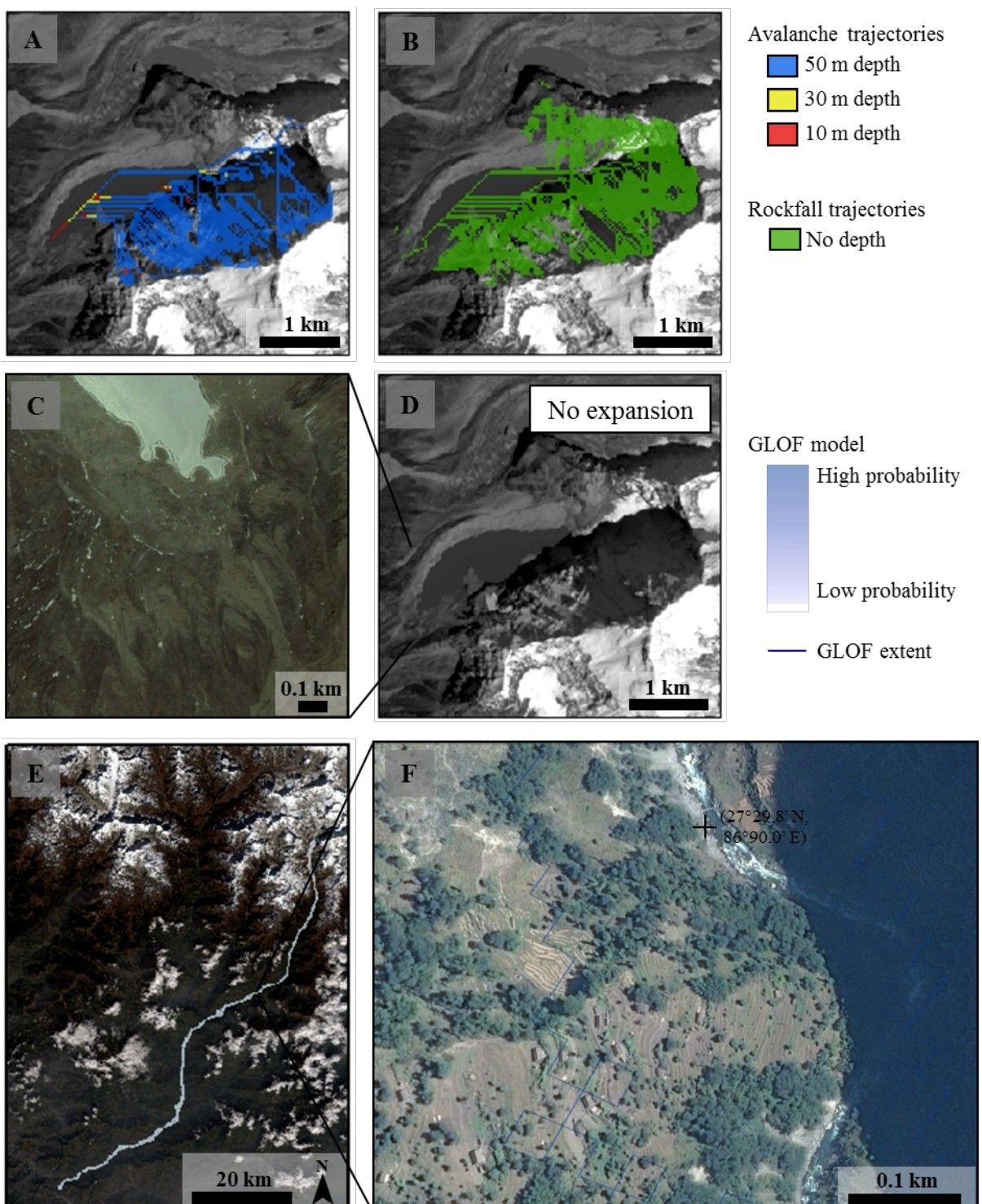

2  Figure S1. Hazards and downstream impact for Chamlang North Tsho: (A) avalanche trajectories,

3  (B) rockfall trajectories (C) lack of ponds on the moraine, (D) future lake expansion, and (E) the

4  extent of MC-LCP GLOF model (F) highlighting the impacts downstream.  Background image

5  (A-E) is Landsat 8 from 30 September 2015 and (F) Google Earth.

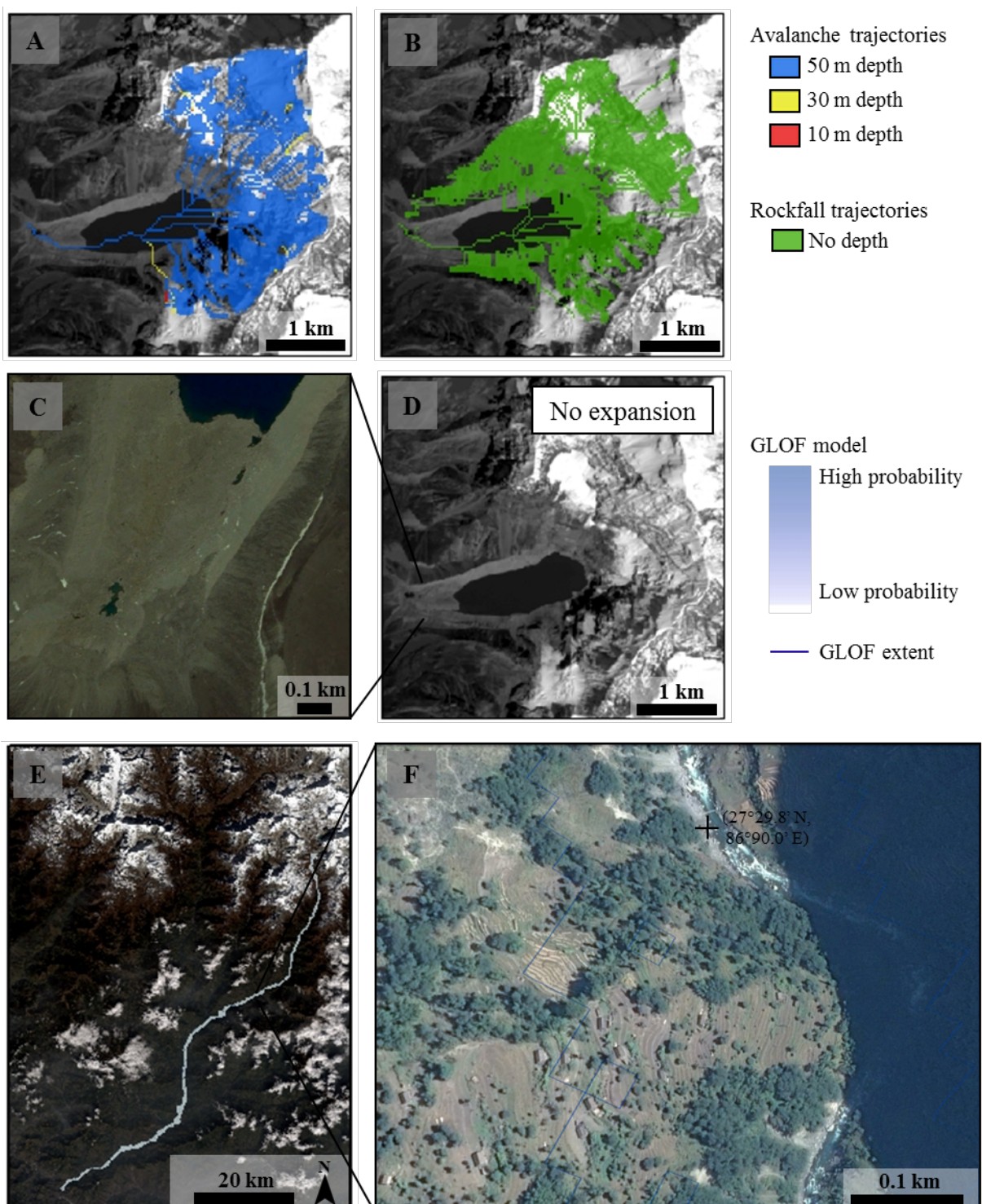

2 Figure S2. Hazards and downstream impact for Chamlang South Tsho: (A) avalanche trajectories,

3 (B) rockfall trajectories (C) ponds on the moraine, (D) future lake expansion, and (E) the extent

4 of MC-LCP GLOF model (F) highlighting the impacts downstream.  Background image (A-E) is

5 Landsat 8 from 30 September 2015 and (F) Google Earth.

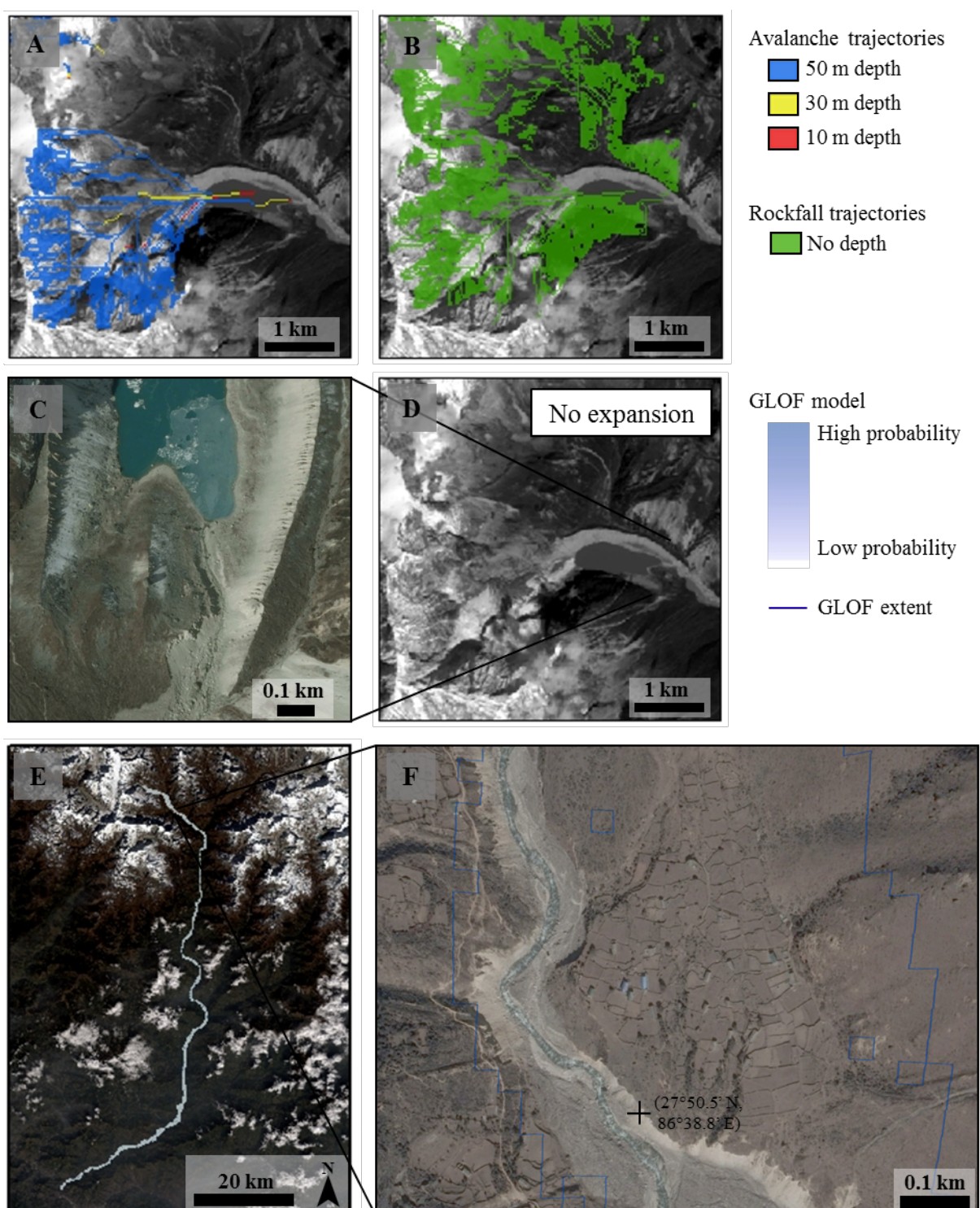

Figure S3. Hazards and downstream impact for Dig Tsho: (A) avalanche trajectories, (B) rockfall trajectories (C) ponds on the moraine, (D) future lake expansion, and (E) the extent of MC-LCP GLOF model (F) highlighting the impacts downstream. Background image (A-E) is Landsat 8 from 30 September 2015 and (F) Google Earth.

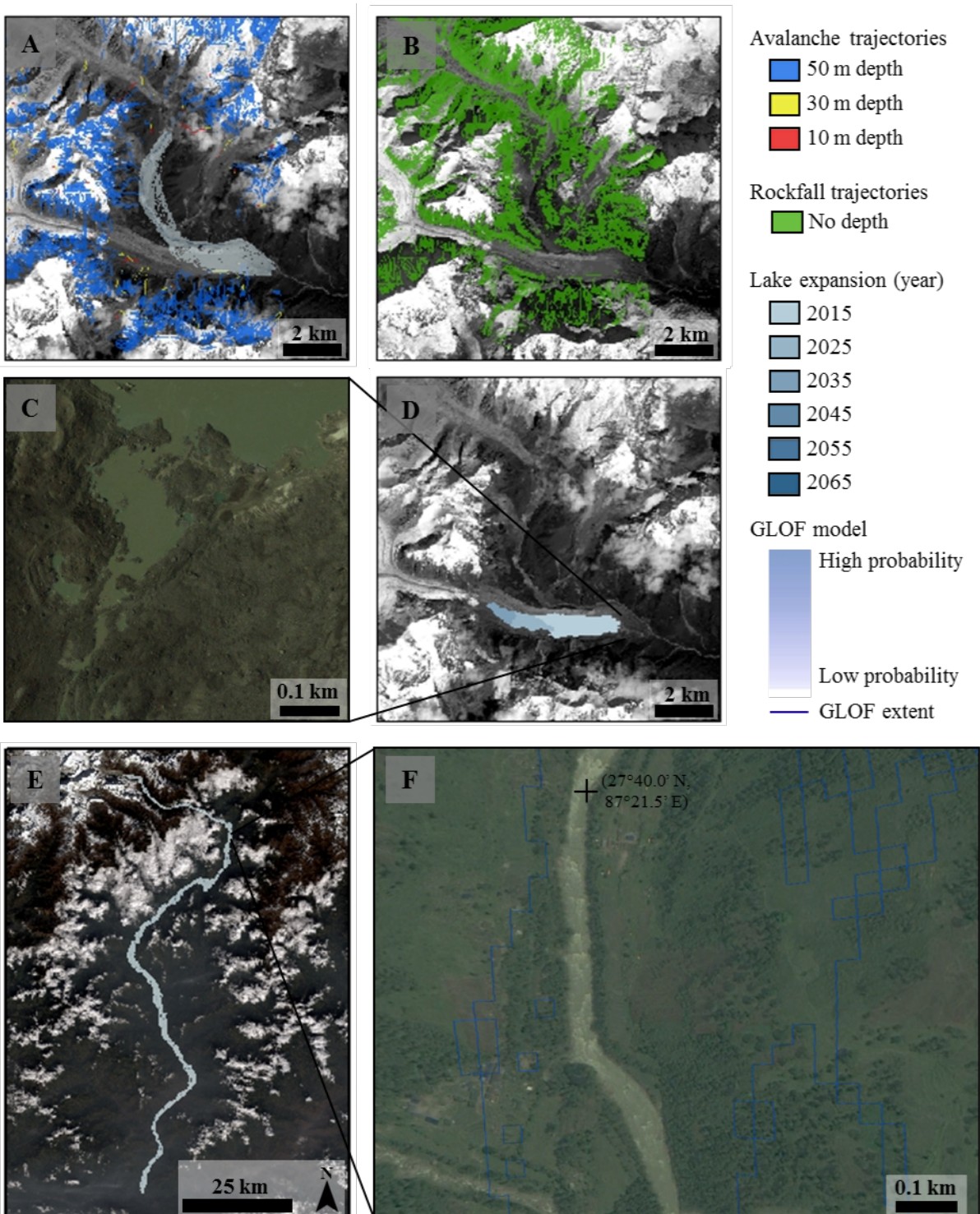

2 Figure S4. Hazards and downstream impact for Lower Barun Tsho: (A) avalanche trajectories

3 and upstream GLOF, (B) rockfall trajectories (C) ponds on the moraine, (D) future lake

4 expansion, and (E) the extent of MC-LCP GLOF model (F) highlighting the impacts downstream.

5 Background image (A-E) is Landsat 8 from 30 September 2015 and (F) Google Earth.

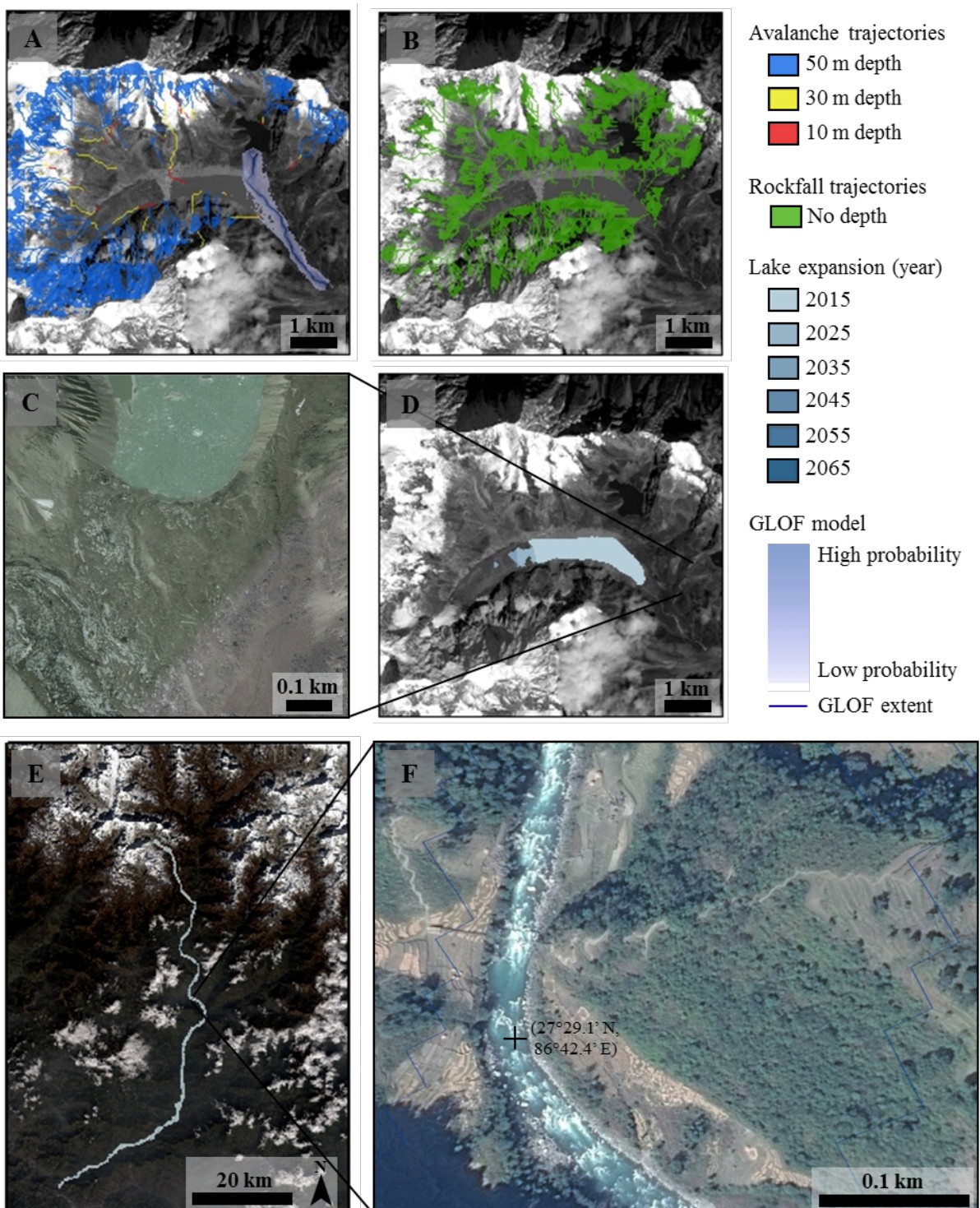

2 Figure S5. Hazards and downstream impact for Lumding Tsho: (A) avalanche trajectories and

3 upstream GLOF, (B) rockfall trajectories (C) ponds on the moraine, (D) future lake expansion,

4 and (E) the extent of MC-LCP GLOF model (F) highlighting the impacts downstream.

5 Background image (A-E) is Landsat 8 from 30 September 2015 and (F) Google Earth.

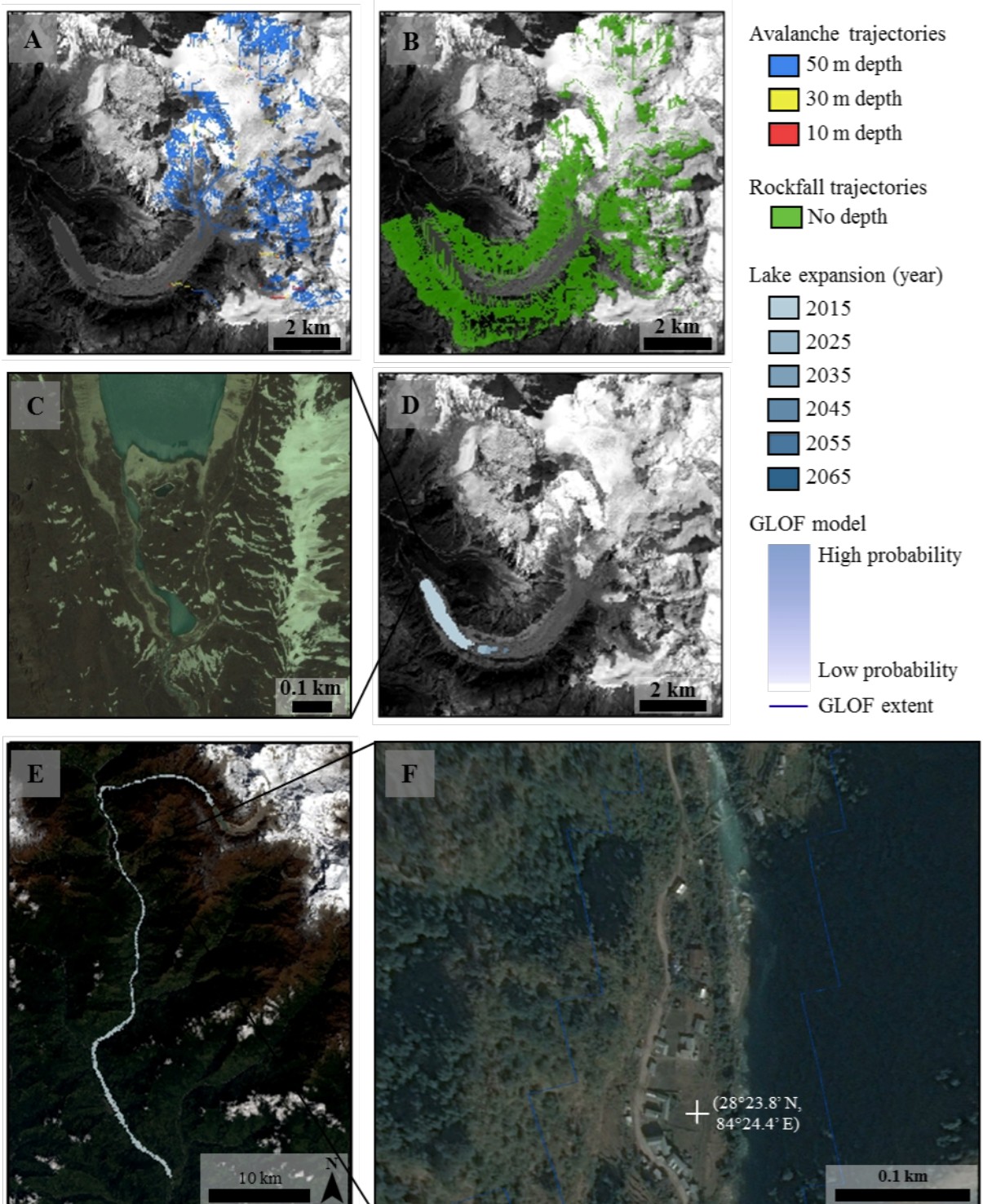

2 Figure S6. Hazards and downstream impact for Thulagi Tsho: (A) avalanche trajectories, (B)

3 rockfall trajectories (C) ponds on the moraine, (D) future lake expansion, and (E) the extent of

4 MC-LCP GLOF model (F) highlighting the impacts downstream. Background image (A-E) is

5 Landsat 8 from 15 November 2015 and (F) Google Earth.

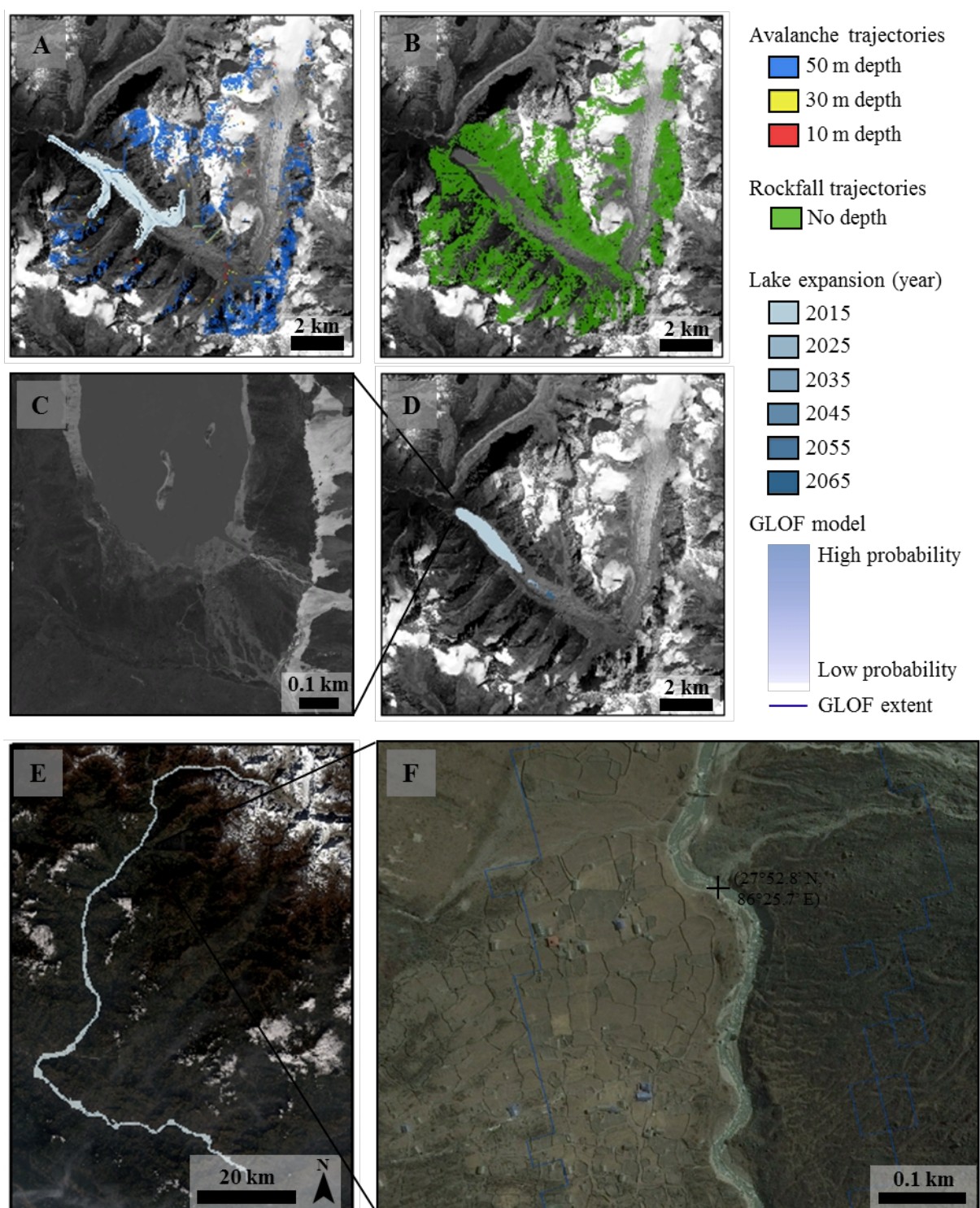

2   Figure S7. Hazards and downstream impact for Tsho Rolpa: (A) avalanche trajectories, (B)

3   rockfall trajectories (C) ponds on the moraine, (D) future lake expansion, and (E) the extent of

4   MC-LCP GLOF model (F) highlighting the impacts downstream.  Background image (A-E) is

5   Landsat 8 from 30 September 2015 and (F) Google Earth.