# Peer review of "A New Remote Hazard and Risk Assessment Framework for Glacial"

_Hydrology and Earth System Sciences, 2016_

## Referee Comment (RC1) · Anonymous Referee #1 · 3 Jun 2016

The authors present an interesting and useful contribution to the increasingly important research concerning techniques and methods for early hazard assessments related to potential outburst floods in large mountain regions. Their pragmatic approach is certainly worth publishing as: (1) adequate discussion of comparable approaches described with sufficient literature citation;(2) developing a new holistic remote sensing based approach which could be promising applied in other mountain regions, e.g., in the north slope of Himalaya;(3) comprehensive state knowledge of the eight glacial lakes in Nepal was summarized followed by propose explanation, which help guide future field campaigns and risk-mitigation strategies. However, after assessing the manuscript the authors submitted, I think it is necessary to work on improvements before it is satisfactory for publication. (1) In 4.1.1 Mass movement trajectories, page 10. Here avalanche only indicated ice avalanche? How do you consider the snow
avalanche? Can the thickness of ice avalanche be largely calculated using the average surface slope of the dangerous glacier (see Wang et al., 2012)? Please discuss it. (2) The DEM data based MC-LCP GLOF model lack the ability to model the different aftermaths caused by flash flood or debris flow. Inundated area by flood or debris flow can bring different catastrophic Consequence, the downstream impact of eight glacial lakes in Nepal Himalaya modeled MC-LCP GLOF model was only resulted from flash flood? (3) In 4.1.6 Downstream Impact, it is better if a Table or Chart was used to illustrate the potential downstream impact classes. (4) In 4.2 Risk classification and management actions, how do you think the changes of permafrost? The permafrost degradation is ubiquitous in Himalaya, it likely makes the glacial lakes more susceptible both in dynamic and self-destructive failure. And the permafrost degradation detected by INSAR was reported recently. (5) How do you obtain socio-economic data (e.g. buildings, agricultural land etc.) the downstream of the eight lakes? Is it cited from ICIMOD, 2011? Or obtained from remote sensing images? (6) In the manuscript, the errors of lake area and change rate were not described. It had better explain the errors in the paper, e.g., explain it at the bottom of Table S7 by note. (7) For preciseness, the yellow line denotes the S.Chamlang Tsho should be added in Figure 1. (8) Please add the detailed source information of images of Figure S1-S7.

---

## Referee Comment (RC2) · Anonymous Referee #2 · 13 Jun 2016

General comments: The contribution of this work is significant, because the authors bring new data about remote area which is hard to access and the methodology is appropriate in general. However the methodology should have its own chapter. In this paper are the methodological aspects incorporated into the chapter 4., which also State of Art. Beside this results are connected in one chapter with discussion. I do not see the reason for this. It will be better to present clearly the results and discuss in a separate chapter new results with already published papers (or to show limits of this methodology).

Specific comments: I have doubts that the most common trigger of GLOFs is mass movement everywhere – there could be regional differences (page 2, line 8-10).

Chapter 4.1.1.: I do not understand why landslides were excluded. There could be

landslides from the inner part of the lateral moraines and due to the glacier retreat new fresh slopes will be prone for sliding. Such waves might be smaller but they could trigger cascade effect – e.g. increase erosion of the dam.

Page 10, line 1: why you set up the upper limit for rockfall prone areas for 60° ?

Page 10, line 1: lateral moraines are not "rockfall prone" areas but "landslide prone" areas ! It is true that they are well developed from morphological point of view but they are not stable. They are too fresh. It is also true that they will not loose large amount of material in one moment but cascade effect could happen (see above).

Chapter 4.1.5: In the list of GLOFs models could be also the HEC-RAS model mentioned (e.g. Klimeš J., Benešová M., Vilímek V., Bouška P., Cochachin A.R. (2014): The reconstruction of a glacial lake outburst flood using HEC-RAS and its significance for future hazard assessments: an example from Lake 513 in the Cordillera Blanca, Peru. Natural Hazards, 71, 3, 1617-1638, 10.1007/s11069-013-0968-4).

Page 19, chapter 5.1. Is Imja Tsho a supraglacial lake? Probably not – it looks like proglacial lake from Google Earth. It is good to use specific names of glacial lakes – more precise.

Page 23, line 14-15: I don believe that the lateral moraine could protect the lake. If you look on Google Earth there is visible a clear contribution of sediments from N-NE (a dejection cone) which already party destroyed the lateral moraine.

Page 41, Fig 5: Is there some reason that GLOFs hazards is in 4 categories and the downstream impact only in 3 classes? If there will be 3 x 3 categories I will suggest the following combinations: H x H = H M x H = H H x M = H

M x M = M M x L = M L x M = M and L x L = L Otherwise my question is why: the combination of H x M = H and M x H results "only" in M ?

---

## Author Comment (AC1) · 21 Jul 2016

**Authors' Response to Reviewers' Comments**

Thanks to both reviewers for their insightful and constructive comments. The following response seeks to address all these comments and detail the subsequent revisions to the text.

**Response to Referee #1 Comments**

(1) In 4.1.1 Mass movement trajectories, page 10. Here avalanche only indicated ice avalanche? How do you consider the snow avalanche? Can the thickness of ice avalanche be largely calculated using the average surface slope of the dangerous glacier (see Wang et al., 2012)? Please discuss it.

> The mass movement trajectories account for both snow and ice avalanches as the band ratio used does not differentiate snow and ice as described in the text (Page 9, Lines 25-27). Therefore, in the text the term "avalanche" is used to describe snow and/or ice avalanches. One limitation of this study is the volume of the avalanche is only based on estimated ice thickness and does not account for accumulated snowfall. We believe this is a relatively minor issue as the assumed ice thickness ranges from $10 - 50$ m, so this large range should account for any variations that would arise from accumulated snowfall.

> Avalanche thickness estimates from Wang et al. (2012) have been assessed to determine the reasonableness of the assumed ice thickness estimates. The estimate is based on a relation between ice thickness, shear stress, surface slope, and a slope factor. The avalanche prone areas have slopes ($\alpha$) that range from 45° to 60°. A range of values for shear stress ($\tau = 100 - 150$ kPa) and slope factor ($k = 0.5 - 0.9$) were used with slopes of 45° and 60° to determine the minimum and maximum ice thickness for these prone areas. The ice thickness was found to range from 15 m ($\alpha = 60°$, $\tau = 100$ kPa, $k = 0.9$) to 48 m ($\alpha = 45°$, $\tau = 150$ kPa, $k = 0.5$), which agrees very well with the assumed ice thickness estimates of $10 - 50$ m. Based on this good agreement and the uncertainty around estimating avalanche thickness, the assumed thickness values will still be used. The text has been revised to include this good agreement as a justification for the assumed values used in this study.

(2) The DEM data based MC-LCP GLOF model lack the ability to model the different aftermaths caused by flash flood or debris flow. Inundated area by flood or debris flow can bring different catastrophic Consequence, the downstream impact of eight glacial lakes in Nepal Himalaya modeled MC-LCP GLOF model was only resulted from flash flood?

> The GLOF model used in this study is a very conservative estimate of GLOF extent. The MSF model has the ability to differentiate between flash flood and debris flows based on the threshold of average slope used to terminate the simulation; however, as discussed in the text (Section 4.1.5), the MSF appeared to severely underestimate the

GLOF extent.  We recommend that debris flows and flash floods be considered in future work where physically-based GLOF models are used, which are more suitable to handle these complex processes.   Text has been added to Section 4.1.5 in the descriptions of the MSF and MC-LCP models to explicitly state that the MSF model can be used to estimate debris flows, while the MC-LCP does not differentiate between the two.

(3) In 4.1.6 Downstream Impact, it is better if a Table or Chart was used to illustrate the potential downstream impact classes.

A new table (Table 4 in the revised paper) has been added that gives a detailed description of each of the four classifications.  It is important to note that in response to the other reviewer's comments, a fourth classification of downstream impacts has been added.

(4) In 4.2 Risk classification and management actions, how do you think the changes of permafrost? The permafrost degradation is ubiquitous in Himalaya, it likely makes the glacial lakes more susceptible both in dynamic and self-destructive failure. And the permafrost degradation detected by INSAR was reported recently.

The reviewer raises a good point that changes in permafrost are likely to have significant impacts in the future stability of the slopes surrounding glacial lakes and hence the likelihood of mass movement entering a lake.  Haeberli et al. (2016) discusses this phenomenon and specifically uses Imja Tsho, one of the lakes in this study, as an example of the future impacts.  Unfortunately, incorporating permafrost degradation and future slope stability into the remote hazard assessment is beyond the scope of this study.  Haeberli et al. (2016) states that critical slope conditions (> 40°) could be combined with critical permafrost thermal conditions (> -1°C), which could be used to assess long-term degradation.  Therefore, it is possible that the use of a 30° threshold to identify prone rock areas may encompass this future degradation.  The citation to Haeberli et al. (2016) discussing the future effects of permafrost degradation has been added to Section 4.2.

(5) How do you obtain socio-economic data (e.g. buildings, agricultural land etc.) the downstream of the eight lakes? Is it cited from ICIMOD, 2011? Or obtained from remote sensing images?

The source of the socio-economic data is described in Section 4.1.6 (Page 16, Lines 14-18).  The buildings were from OpenStreetMap that were then validated and supplemented (when necessary) by the latest Google Earth imagery.  The agricultural land was manually delineated in Google Earth.

(6) In the manuscript, the errors of lake area and change rate were not described. It had better explain the errors in the paper, e.g., explain it at the bottom of Table S7 by note.

The maximum error in the lake area estimates was calculated as the length of the perimeter of the lake multiplied by half the pixel resolution ($\pm$ 15 m). The authors considered reporting the error in the estimates in the supplemental table, but this did not provide very much additional information and made the table crowded. The errors for each image (Table R1) show the error as a percentage of lake area, which ranges from 4.3 to 14.5%. The average error was 10%. The method used to calculate lake error has been added to the text in Section 4.1.2 and the range and average error has been added as a note to Table S7. It is important to note that this is the maximum error.

Table R1. Lake area error shown as a percentage of lake area for each lake area reported in this study

| Year | Lake Area Error (% of lake area) | | | | | | | |
|------|------|------|------|------|------|------|------|------|
| | Imja Tsho | Barun Tsho | Lumding Tsho | Thulagi Tsho | Tsho Rolpa | Chamlang N. Tsho | Chamlang S. Tsho | Dig Tsho |
| 2000 | 9.9 | 8.0 | 10.8 | 12.3 | 9.7 | - | 9.4 | 14.2 |
| 2001 | 10.5 | 8.0 | 10.8 | 12.4 | 9.6 | - | - | - |
| 2002 | 10.1 | 8.5 | 10.7 | 12.1 | 9.9 | - | - | - |
| 2003 | 10.3 | 4.6 | 12.0 | 12.3 | 9.9 | - | - | - |
| 2004 | 10.4 | 4.3 | 10.8 | 12.2 | 10.0 | - | - | - |
| 2005 | 10.2 | 7.6 | 11.3 | 12.1 | 9.4 | - | 9.6 | 14.2 |
| 2006 | 9.3 | 4.3 | 10.5 | 12.2 | 9.6 | - | - | - |
| 2007 | 9.0 | 8.0 | 11.2 | 12.0 | 9.6 | - | - | - |
| 2008 | 8.3 | 7.3 | 10.3 | 12.2 | 9.7 | - | - | - |
| 2009 | 8.0 | 8.9 | 10.1 | 12.1 | 9.2 | - | - | - |
| 2010 | 8.2 | 8.1 | 10.7 | 12.0 | 9.3 | - | 9.7 | 14.5 |
| 2011 | 7.9 | 8.0 | 10.4 | 12.3 | 9.7 | - | - | - |
| 2012 | 8.3 | 7.5 | 10.4 | 12.3 | 9.5 | - | - | - |
| 2013 | 8.2 | 7.8 | 10.6 | 12.5 | 9.6 | - | - | - |
| 2014 | 8.0 | 8.1 | 10.3 | 12.7 | 9.7 | - | - | - |
| 2015 | 7.9 | 7.7 | 10.2 | 12.4 | 9.9 | 11.5 | 9.7 | 14.0 |

The developed method to select the threshold based on a constant width of the lake is meant to reduce the uncertainty associated with comparing lake area over multiple years. The width of the yearly delineations of the lake were all within one tenth of a pixel with respect to the reference image, i.e., the width of any lake was within < 1/10 pixel compared to the width of the reference image. Therefore, the uncertainty associated with the change in area between any two given years should be a maximum of 2/10 of a pixel (6 m for Landsat imagery) multiplied by the perimeter. As opposed to reporting the errors associated with the change in the lake area, which would require a Monte Carlos simulation to propagate the errors through and is

beyond the scope of this study, the standard deviation of the growth rates was reported instead. This was reported in the discussion for all the lakes that were potentially growing (Imja Tsho, Lumding Tsho, Barun Tsho, Thulagi Tsho, and Tsho Rolpa).

(7) For preciseness, the yellow line denotes the S.Chamlang Tsho should be added in Figure 1.

 The yellow line has been added to the figure.

(8) Please add the detailed source information of images of Figure S1-S7.

 This information has been added to Figure S1-S7 and to Figure 7 as well.

**Response to Referee #2 Comments**

General comments: The contribution of this work is significant, because the authors bring new data about remote area which is hard to access and the methodology is appropriate in general. However the methodology should have its own chapter. In this paper are the methodological aspects incorporated into the chapter 4., which also State of Art. Beside this results are connected in one chapter with discussion. I do not see the reason for this. It will be better to present clearly the results and discuss in a separate chapter new results with already published papers (or to show limits of this methodology).

 Section 4.1 is devoted to detailing the methods used with the new hazard and risk framework. The Section Title has been changed to "Remote hazard assessment: Methods" such that this is more explicit.

 The results and discussion have been split into separate sections as the reviewer recommended.

Specific comments: I have doubts that the most common trigger of GLOFs is mass movement everywhere – there could be regional differences (page 2, line 8-10).

 The reviewer is correct that there are regional differences worldwide. The text has been revised to reflect that mass movement entering the lake is the most common triggering event in the Himalaya.

Chapter 4.1.1.: I do not understand why landslides were excluded. There could be

landslides from the inner part of the lateral moraines and due to the glacier retreat new fresh slopes will be prone for sliding. Such waves might be smaller but they could trigger cascade effect – e.g. increase erosion of the dam.

Landslides from the lateral moraines have been observed entering the lake frequently by the authors on previous field expeditions at Imja Tsho. These landslides are very small in size and hence have very little impact on the erosion of the terminal moraine. Additionally, the height of the lateral moraines is typically less than 100 m, such that the when these landslides do enter the lake, they enter the lake with very little energy in comparison to a landslide/rockfall from the side slopes that could be much larger and/or fall from a greater elevation. Furthermore, based on documented GLOFs in the Himalaya, Emmer and Cochachin (2012) report only a single GLOF event that was triggered by a landslide, rockfall, or liquid water. As this paper is focused on using simple models to rapidly assess the hazard of a glacial lake from remote sensing, the authors feel justified in excluded landslides form lateral moraines as they appear unlikely to be the main cause of a GLOF.

Page 10, line 1: why you set up the upper limit for rockfall prone areas for 60°?

The upper limit of the prone areas has been removed and the figures/results have been revised accordingly.

Page 10, line 1: lateral moraines are not "rockfall prone" areas but "landslide prone" areas! It is true that they are well developed from morphological point of view but they are not stable. They are too fresh. It is also true that they will not loose large amount of material in one moment but cascade effect could happen (see above).

In this study, the terms rockfall and landslide are used synonymously to refer to mass movement from any non-glacierized area. A sentence stating this has been added to Section 4.1.1.

Chapter 4.1.5: In the list of GLOFs models could be also the HEC-RAS model mentioned (e.g. Klimeš J., Benešová M., Vilímek V., Bouška P., Cochachin A.R. (2014): The reconstruction of a glacial lake outburst flood using HEC-RAS and its significance for future hazard assessments: an example from Lake 513 in the Cordillera Blanca, Peru. Natural Hazards, 71, 3, 1617-1638, 10.1007/s11069-013-0968-4).

There are many different models that have been used to simulate GLOFs globally. Westoby et al. (2014) was referenced in Section 4.1.5 to summarize and highlight the various models and their applications, which includes HEC-RAS. The only GLOF models that were explicitly discussed in the text were those that have been performed on one of the eight glacial lakes that were studied.

Page 19, chapter 5.1. Is Imja Tsho a supraglacial lake? Probably not – it looks like proglacial lake from Google Earth. It is good to use specific names of glacial lakes – more precise.

> Yes, Imja Tsho and all the other glacial lakes in this study are proglacial lakes. The text in Section 2 has been revised to reflect this fact.

Page 23, line 14-15: I don't believe that the lateral moraine could protect the lake. If you look on Google Earth there is visible a clear contribution of sediments from N-NE (a dejection cone) which already partly destroyed the lateral moraine.

> The text has been revised to read "may or may not protect the lake" as the simplistic GIS-based models used in this study do not account for the erosion required to determine how a lateral moraine may breach during a GLOF. Hence, in this section, the authors highlight the importance of modeling a potential upstream GLOF from Lumding Teng Tsho using more complex models that can account for the erosion of the lateral moraine. It is beyond the scope of this study to determine whether or not the lateral moraine will breach.

Page 41, Fig 5: Is there some reason that GLOFs hazards is in 4 categories and the downstream impact only in 3 classes? If there will be 3 x 3 categories I will suggest the following combinations: H x H = H, M x H = H, H x M = H, M x M = M, M x L = M, L x M = M and L x L = L Otherwise my question is why: the combination of H x M = H and M x H results "only" in M?

> The reviewer brings up an excellent point. Figure 5 has been altered to a 4 x 4 matrix that uses the same combinations as Worni et al. (2013). The fourth classification created for downstream impacts reflects the most damaging case, which includes both potential loss of life without warning (buildings/lodges) and potential damage to costly infrastructure/projects (e.g., hydropower). The text in Section 4.1.6 has been altered to include this change. Furthermore, the downstream impacts and risk classification for many of the glacial lakes in this study changes, which is also reflected in the Section 5 (results) and Section 6 (discussion).

References
Haeberli, W., Shaub, Y., and Huggel, C.: Increasing risks related to landslides from degrading permafrost into new lakes in de-glaciating mountain ranges, *Geomorphology*, 2016, http://dx.doi.org/10.1016/j.geomorph.2016.02.009